# Absolute radiance calibration in the UV and visible spectral range using atmospheric observations during twilight

Thomas Wagner, and Jānis Puķīte

Satellite Remote Sensing Group, Max Planck Institute for Chemistry, Mainz, Germany

*Correspondence to*: Thomas Wagner (thomas.wagner@mpic.de)

**Abstract.** We present an improved radiance calibration method for UV / visible spectroscopic instrument s with a narrow field of view (up to a few degrees) based on the calibration method by Wagner et al. (2015). The updated method uses only measurements during the twilight period instead of several hours as for the original method. The calibration is based on the comparison of measurements and simulations of the radiance of zenith-scattered sun light. The main advantage of our method compared to radiance calibration methods in the laboratory is that the calibration can be directly applied in the field.
This allows routine radiance calibrations whenever the sky is clear during twilight. The calibration can also be performed retrospectively, and will thus be applicable for the large number of existing data sets. Also, potential changes of the instrument properties during transport from the laboratory to the field are avoided. The new version of the calibration method presented here has two main advantages: First, the required measurement period can be rather short (only a few minutes during twilight for cloud-free conditions). Second, even without knowledge of the aerosol optical depth, the errors
of the calibration method are rather small, especially in the UV spectral range where they range from about 4 % at 340 nm to 8 % at 420 nm. If the AOD is known, the uncertainties are even smaller (about 3 % at 340 nm to 4 % at 420 nm ). For visible wavelengths, good accuracy is only obtained if the AOD is approximately known with uncertainties from about 4 % at 420 nm to 10 % between about 550 and 700 nm (generally the AOD is nevertheless smaller in the visible than in the UV spectral range). One shortcoming of the method is that it is not possible to determine the AOD exactly at the time of the (twilight)
measurements, because AOD observations from sun photometer measurements or the MAX-DOAS measurements are usually not meaningful for such high SZA. But the related uncertainty can be minimised by repeating the radiance calibrations during the twilight periods of several days.

## 1. Introduction


Measurements of the atmospheric radiance are important for many applications, e.g. atmospheric remote sensing, studies of atmospheric photochemistry, optimisation of the energy yield of photovoltaic cells, the classification of sky conditions, determination of absorbing properties of aerosols, or the quantification of biologically relevant UV doses (for more details see e.g. Riechelmann et al., 2013, Wagner et al., 2015). For some of these applications (e.g. atmospheric photochemistry or
the energy yield of photovoltaic cells), the relative uncertainties of the radiance measurements will cause similar relative uncertainties of the derived quantities. For other quantities, however, the relationship can be non-linear: for example, a 5% error of the measured radiance can lead to errors of the derived aerosol single scattering albedo of up to 10% (Dubovik et al., 2000) or to a change of the number of detected optically thick clouds (Wagner et al., 2014; 2016) of up to 15%.

Usually, measurements of the spectral radiance are calibrated in the laboratory using calibrated light sources (e.g. Pissulla et
al., 2009; Yu et al., 2014, Niedzwiedz et al., 2021). Uncertainties of the calibration procedures typically range from 3 to 10 % (Wuttke et al., 2006; Pissulla et al., 2009). However, in comparison studies differences between individual instruments up to 30 % have been reported (Pissulla et al., 2009). One particular problem arises from the fact that during transport of the instrument from the laboratory to the field the instrument properties might change.

In this study we build on a recently introduced radiance calibration method using atmospheric radiance measurements in zenith direction (Wagner et al., 2015). In the original study, the radiance calibration was performed by the comparison of measured and simulated zenith radiances under favorable conditions (cloud-free sky, stable aerosol conditions). But in contrast to rather long measurement periods needed with the original method (several hours), the updated method can be applied for much shorter periods (usually a few minutes) during twilight (thus measurements at lower SZAs are not required in the new method). The original and the updated method are based on the comparison of atmospheric radiance measurements to atmospheric radiative transfer simulations for a cloud-free atmosphere and low and stable aerosol abundance. One important advantage compared to calibration measurements in the laboratory is that the calibration can be applied directly in the field without the need to transport the instrument to the laboratory and back. Compared to the original method (Wagner et al., 2015), the updated method also requires less specific and less stable atmospheric conditions, especially with respect to the aerosol load. Moreover, during twilight, aerosols have a rather weak effect on the observed radiance (Fig. 1a), especially in the UV. The small influence of atmospheric aerosols on the zenith scattered solar radiance in the UV can be explained by the fact that during twilight the altitude, from which the solar light is scattered into the instrument, increases with decreasing wavelengths because of the increased probability of Rayleigh scattering (Fig. 2). Thus, scattering by aerosols (which usually reside close to the surface) does not substantially increase the observed radiance (as will be shown later in section 2.3, the phase function and the single scattering albedo might still have a relatively strong effect). Another advantage of the new method is that the pointing accuracy is much less important compared to the original method. In Fig. 1b measured zenith radiances from MAX-DOAS measurements during September and October 2022 are shown. The light blue dots represent all measurement conditions, whereas the dark blue dots represent only clear sky conditions (4.5 days). Like for the simulated radiances, also the measured zenith radiances for clear sky conditions show only small variations, especially during twilight. Between 88° and 90°, the deviation from a fitted polynomial (within the SZA interval 85° to 93°) is about ±1.5 % and ±4 % at 345 nm at 445 nm, respectively (Fig. 1c). During these clear days, the AOD measured by a sun photometer at the same location (https://aeronet.gsfc.nasa.gov/new_web/index.html) varied between 0.08 and 0.18 at 340 nm and between 0.05 and 0.13 at 440 nm.

In this study we explore the applicability of the new method mainly based on radiative transfer simulations, in particular we determine the optimum SZA range. We also quantify the remaining uncertainties caused by incomplete knowledge of the atmospheric state, the position of the instrument, the surface albedo, and possible errors of the measurements and radiative transfer simulations.

The paper is structured as follows: In section 2 the radiative transfer simulations are described and sensitivity studies on the influence of atmospheric aerosols on the zenith scattered sun light during twilight are presented. In section 3 the new method is applied to measurements and the results are compared to the calibration results of the original study. Section 4 provides a summary and conclusions.

**2 Radiative transfer simulations**

In the first part of this section the radiative transfer model and the atmospheric settings used for the simulations are introduced. In the following parts the effects of aerosols and of other atmospheric and measurement parameters on the simulated zenith scattered radiances are explored. Based on these results the optimum SZA range for the application of the method is determined and the remaining uncertainties are quantified.

**2.1 Radiative transfer model and atmospheric scenarios**

In the original study, the radiative transfer model MCARTIM-3 (Deutschmann et al., 2011) was used. MCARTIM is a a full spherical Monte-Carlo model, for which, however, under twilight conditions the noise can become rather high. Thus, in this study we decided to use the radiative transfer model SCIATRAN (version 3.8.11, Rozanov et al., 2017; Mei et al., 2023, https://www.iup.uni-bremen.de/sciatran/) in order to minimise the uncertainties due to noise of the simulations and the computational effort. We use SCIATRAN in sphericity mode and considered polarisation and refraction (in the ray tracing calculations). For the sensitivity studies (section 2), also rotational Raman scattering is considered. A Lambertian surface albedo was used. For the conditions of this paper, the zenith scattered radiances during twilight, simulated by both models agree within ±1%. Other radiative transfer models with similar capabilities could also be used for the radiance calibration. For the radiative transfer simulations the vertical resolution was 50 m below 2 km, about 1 km between 2 km and 18 km, and up to a few kilometers above. Further settings of the simulations are summarised in Table 1. Zenith radiances are simulated for the SZA range from 85° to 93° for the wavelength range 340 nm to 700 nm (in steps of 40 nm).

### 2.2 Dependence on the AOD

The dependence of the simulated radiance on the AOD for the standard settings (Table 1) is shown in Fig. 3 and Fig. A1 in the appendix. The effect of the AOD increases towards longer wavelengths with deviations of about 7 % and 30 % at 340 nm and 700 nm, respectively (for an SZA of 90° and an AOD of 0.5 compared to an AOD of 0). This is a result of the increase of the penetration depth of the direct sun light into the atmosphere towards longer wavelengths. Thus scattering by aerosols (in addition to Rayleigh scattering) becomes increasingly important towards longer wavelengths. Here it is worth noting that in general the AOD decreases with increasing wavelength. Another important finding is that for wavelengths below about 500nm, the smallest effects of aerosols are found for SZA around 89° to 90°. This is also confirmed by measurements (see Fig. A2 in the appendix). For longer wavelengths, the smallest effects are found for slightly larger SZA (up to 92°).

### 2.3 Uncertainties caused by other atmospheric and surface properties

In addition to the dependence on the AOD, in this section, the effect of other important properties is investigated. Here assumptions about the variability range of the different quantities were made (see table 2) which should be representative for typical atmospheric conditions. In specific cases, some of these properties might be outside the assumed ranges (e.g. for events with desert dust or biomass burning plumes). Such extreme cases should be avoided for the application of the calibration method. In Fig. 4 and Fig. A3 the ratios of the simulated radiances compared to those of the corresponding standard scenarios (see Table 1) for two selected AODs (0.1 and 0.3) are shown. These AODs were chosen, because they represent typical conditions in most parts of the world. It is again found that overall the uncertainties increase towards longer wavelengths with deviations of about ± 3 % and ± 8 % at 340 nm and 700 nm, respectively (for an SZA of 90°). For short wavelengths, the effect of stratospheric aerosols and variations of the aerosol single scattering albedo have the strongest effect on the simulated radiances. Towards longer wavelengths, the effects of the aerosol phase function and the ozone absorption (Chappuis band) becomes increasingly important. From these findings it is concluded that for the radiative transfer simulations, the correct ozone total column for the day(s) of the measurements should be used (especially for wavelengths between about 500m and 650 nm). Overall, again, for SZA around 89° to 90° the smallest deviations are found.

### 2.4 Uncertainties related to errors of the instrument properties or the chosen options of the radiative transfer simulations

In this section the effect of the altitude of the instrument, its elevation angle calibration and the chosen options of the radiative transfer model (consideration of polarisation, Raman scattering, or atmospheric refraction) are investigated. Of course, the altitude of the instrument is usually well known and could be exactly considered in the radiative transfer simulations. However, if for simplicity the radiances for the standard scenario given in table 3 were used, it will be useful to know, how strongly the altitude of the instrument affects the measured radiances. In Fig. 5 and Fig. A4 the ratios of the simulated radiances for these modifications compared to the radiances for the standard settings are shown. At short wavelengths, the neglect of polarisation in the radiative transfer simulations has the strongest effect (up to 10%). At longer wavelengths, a wrong altitude of the instrument and the neglect of refraction become especially important. From these findings it is concluded that polarisation and refraction have to be taken into account for the radiative transfer simulations. For wavelengths above about 500 nm, also the exact altitude of the measurement site should be used. Then the remaining uncertainties range from about 1% at 340 nm to 3% at 700 nm, repectively. The deviations due to the neglect of Raman scattering in the radiative transfer simulations for short wavelengths is caused by the increased probability for multiple scattering and is related to the specific location of the chosen wavelength. At 340 nm the sun spectrum has a local maximum, whereas at 380 nm and 420 it has a local minimum (Fraunhofer line).

**2.5 Quantification of the uncertainties**

This section summarises the different uncertainties investigated in the subsections before. The resulting errors are quantified for the SZA range from 89° to 90°, for which overall the smallest uncertainties were found.

In the first subsection (2.5.1) the errors are quantified for situations when the AOD is known, e.g. from sun photometer observations (Volz, 1959; Tanre et al., 1988; Kaufman et al., 1994; Holben et al., 1998, and references therein) or MAX-DOAS measurements themselves (e.g. Hönninger et al., 2004; Wagner et al., 2004, Friess et al., 2006; Irie et al., 2008; Clémer et al., 2010, and references therein). Again, two scenarios, one with an AOD of 0.1±0.05, and another with an AOD of 0.3 ± 0.05 were chosen. To account for possible temporal changes of the AOD (MAX-DOAS and AERONET inversions are usually restricted to SZA < 80°), the radiance calibration measurements might be carried out during the twilight period on several succeeding days.

In the second subsection (2.5.2) larger uncertainties for the AOD are assumed (0.25±0.125). This case represents typical aerosol loads for most parts of the globe (except very polluted locations) and might still allow useful radiance calibrations in the UV for situations when the AOD is unknown.

The uncertainties are quantified by variations of the corresponding input parameters as summarised in tables 1 and 2. Note that the deviations caused by the neglect of polarisation and atmospheric refraction are not included in the calculation of the total error, because it is assumed that for the application of our method only radiative transfer simulations are used, which consider both options.

**2.5.1 Uncertainties if AOD is known**

Fig. 6 summarises the individual and total uncertainties for the cases when the AOD is known within ±0.05 (for the specific assumptions see table 2). The sub figures show the error budgets for AODs of 0.1 and 0.3, which represent typical aerosol abundances. The assumptions for the individual error sources (see table 2) are a little bit arbitrary but should represent realistic measurement conditions. Users of the method should check whether these assumptions are realistic for their measurements (especially the total ozone column) and could adjust them if needed. The total error is calculated from the individual errors by assuming that all errors are independent (excluding the effects of polarisation and atmospheric refraction, because only radiative transfer models considering both options should be used). For both scenarios, with the

assumptions made in table 2, the remaining dominant error source for almost all cases is the uncertainty of the AOD. Overall the resulting total uncertainties are still rather small ranging from ±3% for 340nm (AOD = 0.1) to about ±10% for 700nm (AOD = 0.3). Here it should be taken into account that usually the AOD decreases towards longer wavelengths. Thus the total errors for high AOD and long wavelengths probably overestimate the uncertainties for typical scenarios.


### 2.5.2 Uncertainties if AOD is not known

Fig. 7 summarises the individual and total uncertainties for typical cases when the AOD is not known. For the simulations an AOD of 0.25±0.125 was assumed. Again, the dominant error source (execpt the effect of polarisation and refraction) is the

uncertainty of the AOD, and the uncertainties are much larger than for the two cases with known AOD discussed above. However, for the UV radiances the uncertainties are still rather small, about ±4% at 340nm and ±8% at 420nm.

### 3 Application to measurements and comparison to results from previous calibration method

The validation of the method using measured data is difficult, because without dedicated campaigns, reference data sets at the same location as the DOAS measurements are usually not available. Thus, like in the original study (Wagner et al., 2015), we apply the method to the MAX-DOAS measurements made with the MPIC instrument during the CINDI (I) campaign in Cabauw, the Netherlands, during the morning of 24 June 2009 (Piters et al., 2012). This procedure enables a direct comparison of the calibration results of the original and refined method. The MPIC MAX-DOAS instrument is a so-

called Mini-MAX-DOAS instrument which covers the spectral range from 312 to 458 nm with a spectral resolution between 0.45 and 0.8. Its field of view is about 1.2°. The spectral characteristics of the instrument were exactly taken into account in the calibration procedure as described in detail in Wagner et al. (2015).

The calibration factors are derived from the comparison of the measured radiances (in ‚counts' per second) to the simulated radiances (in $Wm^{-2}nm^{-1}sr^{-1}$). For the simulations, the settings for the standard scenario were selected (see Table 1), but also

atmospheric refraction was considered.

For the measurements on 24 June 2009 no zenith measurements were conducted exactly at SZAs of 89° and 90°, because of the rather long elevation sequences (in addition to the zenith view, one elevation sequence included also 11 measurements in non-zenith direction). However, zenith measurements were taken close to those SZAs, i.e. at SZAs of 88.6° and 90.2°. We linearly interpolated the radiances of the measurements at both SZAs and then compared them to the simulation results of

SZAs of 89° and 90° (see Fig. 8) to obtain the calibration factors. The simulated normalised radiances for selected wavelengths and AODs for SZA of 89° and 90° are given in Table 3.

The derived calibration factors are determined for two assumptions:

a) using the simulation results for an AOD of 0.25, assuming that no information on the AOD is available (see section 2.5.2).

b) using the simulation results for the wavelength-dependent AOD derived from the simultaneous sun photometer

observations (Wagner et al., 2015).

The ratios of the calibration factors for these assumptions versus the calibration factors from the original study are shown in Fig. 9. Because of the strong ozone absorption at high SZA, no reasonable calibration results with the refined method for wavelengths < 335 could be obtained. Overall slightly smaller calibration factors (between 2% and 10%) are obtained with the new method (red curve, for the case that the AOD is known) compared to the original method. This means that the

calibrated spectra will have slightly higher radiances. The deviations are still within the uncertainty estimates of the original and new method (old method: ~7%, new method: about 3% to 5% between 335 and 455 nm if the AOD is known). Interestingly, for these measurements, only slightly worse calibration results are obtained if no knowledge about the exact

AOD is available (blue curve). This rather good agreement is probably caused by the fact that the true AODs (between ~0.05 and 0.2, depending on wavelength, see Wagner et al., 2015) are close to the assumed AOD of 0.25).

Part of the differences might be attributed to the following reasons:

a) due to the inclusion of many non-zenith angles in the elevation sequences, the SZA range between 89° and 90° is not well covered leading to interpolation effects, because only a linear interpolation was used. These interpolation effects might cause deviations up to about 2%. In future applications, only zenith measurements should be performed for SZA>89° to better capture the SZA dependence. It should be noted that for state of the art measurements, this is already mostly implemented as

standard measurement routine.

b) The uncertainties caused by the electronic offset and dark current will be higher for measurements during twilight compared to measurements at higher SZA, which determined the calibration factors of the original method. Based on a blind region of the detector we estimated the error caused by possible wrong electronic offset and dark current correction to be <1%.

c) For smaller SZA (like used in the original study) the effect of uncertainties in the scattering phase function is larger than for SZA close to 90°. And this source of uncertainty increases with wavelength. This might at least partly explain the increasing deviation between both calibrations with increasing wavelengths.

Also slight deviations (1% to 3%) between the calibration factors of the new method using either the simulation results of this study (using SCIATRAN) or the simulation results of the original study (using MCARTIM) are found. Part of these

deviations are caused by larger interpolation errors using the old simulation results, because they were made in steps of 2° SZA (compared to 1° in this study). The direct comparison of the results of both radiative transfer models for the considered viewing geometry and atmospheric scenario yields deviations <1%.

### 3.1 Comparison to independent measurements


Figure 10 presents a comparison of the calibrated radiance spectrum measured on 24 June 2009 at 6:54 at a SZA=61° (blue) and a radiance spectrum measured in zenith direction under similar atmospheric conditions (clear sky, SZA = 62°) on 2 May 2007 in Hanover, Germany. We selected this measurement for comparison, because we found no better suited example for the validation of our method in the scientific literature. This reference spectrum was measured by an instrument specifically

designed for atmospheric radiance measurements (Wuttke et al., 2006; Seckmeyer et al., 2009), and was calibrated using a calibration light source. The instrument took part in international comparison studies and was shown to comply with NDSC Standards (Wuttke et al., 2006). The sun-earth distances were quite similar for both measurements. Although the observation geometries and atmospheric conditions are similar for both measurements, still slight deviations can be expected because of the slightly different atmospheric aerosol load. Unfortunately, there was no sun photometer observation available directly at

the Hannover measurement site, but from the AERONET station in Hamburg (about 130 km north of Hannover) a slightly lower AOD compared to Cabauw was found (0.13 compared to 0.17 at 360 nm; 0.10 compared to 0.12 at 440 nm). This difference could explain about 2% to 3% higher radiances in Cabauw compared to Hannover. Note that the measurement in Hannover was scaled by a factor of 0.985 to account for the effect of the slightly different viewing geometries (exact zenith viewand SZA of 62°, compared to 85° elevation angle and SZA of 61° of our measurement, see Wagner et al., 2015). The

bottom of Fig. 10 shows the ratio of both measurements (after the radiances were averaged over intervals of 10 nm). Overall, good agreement is found with the measurements at Cabauw on average about 5% higher. About half of this difference can be attributed to the different aerosol loads as described above. Part of the deviations (especially for the high frequency structures) are probably also related to the fact that the values of the reference spectrum from Hannover were manually extracted from the figure in Seckmeyer et al. (2009), because the spectral data were not available. .


**4 Summary and conclusions**

In this study we presented an improved radiance calibration method compared to our previous study (Wagner et al., 2015). The updated method uses only a short period of measurements (during twilight), whereas in the original method a much longer period (a few hours) on a day with stable aerosol conditions was required. The updated method has two main advantages: First, because of the much shorter measurement period, the method can be used on all days with cloud-free conditions during sunrise and sunset. Second, even without knowledge about the aerosol load, the errors of the calibration method are rather small, especially in the UV spectral range (ranging from about 4% at 340 nm to 8% at 420nm). If the AOD is known, the uncertainties are even smaller (ranging from about 3% at 340 nm to 4% at 420nm). For larger wavelengths, good accuracy is only obtained if the AOD is known (ranging from about 4% at 420 nm to 10% between about 550 and 700 nm). Here it should be noted that usually the AOD in the visible spectral range is systematically smaller compared to the UV.

Another important advantage is that the new calibration method can be applied retrospectively to the large number of existing zenith-sky DOAS measurements.

One disadvantage of the method is that it is not possible to determine the AOD exactly for the time of the (twilight) measurements, because AOD observations from sun photometer measurements or the MAX-DOAS measurements are only possible for SZA smaller than about 80 to 85°. Thus the AOD measured by these methods might differ from the AOD during twilight. One possibility to minimise the related uncertainty is to carry out the radiance calibration during the twilight periods of several days. Then the errors caused by variations of the AOD might largely cancel out. By comparing the results from several days, also the potential effect of clouds far away from the measurement site (but still in the path of the direct sun light) could be identified and contaminated measurements could be removed. Another limitation of the method is that especially for situations with enhanced AOD (see results for AOD of 0.3 in Fig. 4 and Fig. A3) the aerosol properties (phase function and single scattering albedo) can have a relatively strong effect. Such situations (e.g. desert dust events or biomass burning plumes) should be excluded from the application of the calibration technique.

A few more aspects should be mentioned: first, the variation of the earth-sun distance should be taken into account (as was done in this and the original study, see Wagner et al., 2015). If this effect is neglected errors up to about 3.2 % can arise. Second, care should be taken for the exact calculation of the SZA. Especially during twilight, small errors of the computer time and/or the latitude/longitude settings can lead to considerable errors of the SZA calculation. Third, care should also be taken that the saturation level of the detector during twilight is similar to that of typical measurements (at smaller SZA). This was the case for the measurements used in this study. Otherwise, non-linearities of the detector and/or the read-out electronics could lead to systematic errors. Fourth, for a meaningful comparison to the twilight measurements, atmospheric refraction and polarisation has to be taken into account in the radiative transfer simulations. Especially for wavelengths betwwen about 500nm and 650 nm, the total ozone column used in the radiative transfer should match the true ozone column during the measurements.

The main advantage of our method compared to radiance calibration methods in the laboratory is that the calibration can be applied directly in the field. This allows routine radiance calibrations whenever the sky is cloud-free during twilight. Also, potential changes of the instrument properties during transport from the laboratory to the field are avoided.

**Author contributions**

TW developed the idea and wrote the paper. JP performed the radiative transfer simulations and provided valuable input to the interpretation of the results.

**Acknowledgements**


We want to thank the organisers of the Cabauw Intercomparison Campaign of Nitrogen Dioxide measuring Instruments (CINDI) in summer 2009 (https://www.knmi.nl/kennis-en-datacentrum/project/cindi), especially Ankie Piters and Marc Kroon. We thank J. S. (Bas) Henzing and his staff for their effort in establishing and maintaining the Cabauw AERONET site used in this investigation. The radiance measurements at Hanover (Fig. 9) were copied from a publication by Seckmeyer

et al. (2009). We used the Radiative Transfer model SCIATRAN version 3.8.11 developed at University of Bremen (https://www.iup.uni-bremen.de/sciatran/). The measurements used to create Fig. A2 were conducted by Manish Sharma at Sharda University in Greater Nioda, India.

**Competing interests**


Some authors are members of the editorial board of the journal AMT. The peer-review process was guided by an independent editor, and the authors have also no other competing interests to declare.

**Author contributions**


Thomas Wagner initiated the study and wrote the manuscript. Jānis Puķīte performed the radiative transfer simulations and provided valuable input to the study.

**Data availability**


The data are available from the authors upon request.

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

**Tables**

**Table 1 Settings used for the radiative transfer simulations**

| Parameter | Standard scenario | Variations |
|---|---|---|
| polarisation | yes | no |
| Atmospheric refraction | yes | no |
| temperature profile | US standard atmosphere | +20 K for whole profile |
| pressure profile | US standard atmosphere | -2 % relative change for whole profile |
| $O_3$ profile | US standard atmosphere (345 DU) | +20 % relative change for whole profile |
| surface albedo | 0.05 | 0.03, 0.1 |
| Surface altitude | 0 m | 1000 m* |
| elevation angle | 90° | 88° (rel. Azimuth angle: 0, 90°) |
| Raman scattering** | no | yes |
| AOD | 0.1, 0.3 | 0, 0.02, 0.05, 0.1, 0.2, 0.3, 0.5, 1 |
| aerosol layer height | 1000 m | 200m, 500m, 2000m |
| single scattering albedo | 0.95 | 0.9, 1.0 |
| phase function | urban | Marine, biomass burning |
| stratospheric aerosols*** | yes | no |

*the aerosol layer is also shifted by 1000m

**For the investigation of the effect of Raman scattering the simulated spectra are convoluted with a FWHM of 0.6nm and then the average of the radiance for +/-0.25 nm around the selected wavelength is taken

***AOD of 0.012 between 18 and 33km, see Wagner et al. (2021)

**Table 2 Assumptions made for the quantification of different sources of uncertainty**

| Quantity | Tested variation | Weighting of the simulation results to obtain the total radiance error displayed in Figs. 6 and 7 |
|---|---|---|
| Ozone | original ozone profile or profile increased by 20% | half difference between both simulations |
| Temperature | original temperature profile or profile increased by 20K | half difference between both simulations |
| Pressure | original pressure profile or profile decreased by 2% | half difference between both simulations |
| Surface albedo | surface albedo 0.03 or 0.1 | half difference between both simulations |
| AOD | case 1: AOD = 0.1 ± 0.05<br>case 2: AOD = 0.3 ± 0.05<br>case 3: AOD = 0.25± 0.125 | difference between the respective high and low AOD cases |
| Aerosol layer height | layer height 200m or 2000m | half difference between both simulations |
| Aerosol phase function | biomass burning and marine aerosols | half difference between both simulations |
| Aerosol single scattering albedo (SSA) | SSA of 0.9 and SSA of 1.0 | half difference between both simulations |
| Stratospheric aerosols | scenarios with or without stratospheric aerosols | half difference between both simulations |
| Polarisation | simulations with or without polarisation | not included |
| Atmospheric refraction | simulations with or without refraction | not included |
| Raman scattering | simulations with or without Raman scattering | difference between both simulations |
| Instrument elevation | simulations at sea level or 1 km altitude | 10% of the difference between both simulations |
| Pointing accuracy in the plane towards the sun | elevation angles of 88° and 90° | difference between both simulations |
| Pointing accuracy perpendicular to the plane towards the sun | elevation angles of 88° and 90° | difference between both simulations |




**Table 3** Zenith scattered normalised radiances (simulated radiance divided by the solar irradiance) at SZA of 89° and 90° for different wavelenghts, AODs and for the standard settings (see table 1). For the simulations, polarisation and atmospheric refraction were taken into account. The total ozone column was 345 DU.

| Wavelength [nm] | SZA = 89° AOD = 0.1 | AOD = 0.2 | AOD = 0.3 | AOD = 0.5 | AOD = 0.7 | AOD = 1 | SZA = 90° AOD = 0.1 | AOD = 0.2 | AOD = 0.3 | AOD = 0.5 | AOD = 0.7 | AOD = 1 |
|---|---|---|---|---|---|---|---|---|---|---|---|---|
| 340 | 0.00238 | 0.00242 | 0.00245 | 0.00249 | 0.00251 | 0.00252 | 0.00154 | 0.00156 | 0.00158 | 0.00161 | 0.00162 | 0.00162 |
| 350 | 0.00265 | 0.00269 | 0.00273 | 0.00279 | 0.00283 | 0.00284 | 0.00173 | 0.00176 | 0.00179 | 0.00183 | 0.00185 | 0.00186 |
| 360 | 0.00267 | 0.00272 | 0.00277 | 0.00283 | 0.00288 | 0.00291 | 0.00175 | 0.00179 | 0.00182 | 0.00186 | 0.00189 | 0.00191 |
| 370 | 0.00269 | 0.00275 | 0.00280 | 0.00288 | 0.00293 | 0.00297 | 0.00177 | 0.00181 | 0.00184 | 0.00190 | 0.00193 | 0.00195 |
| 380 | 0.00269 | 0.00276 | 0.00282 | 0.00290 | 0.00296 | 0.00301 | 0.00178 | 0.00182 | 0.00186 | 0.00192 | 0.00196 | 0.00199 |
| 390 | 0.00270 | 0.00277 | 0.00283 | 0.00293 | 0.00299 | 0.00305 | 0.00178 | 0.00183 | 0.00187 | 0.00194 | 0.00198 | 0.00202 |
| 400 | 0.00270 | 0.00278 | 0.00284 | 0.00295 | 0.00302 | 0.00308 | 0.00179 | 0.00184 | 0.00189 | 0.00196 | 0.00201 | 0.00205 |
| 410 | 0.00270 | 0.00278 | 0.00285 | 0.00296 | 0.00304 | 0.00311 | 0.00179 | 0.00185 | 0.00190 | 0.00197 | 0.00202 | 0.00207 |
| 420 | 0.00270 | 0.00278 | 0.00286 | 0.00298 | 0.00306 | 0.00314 | 0.00180 | 0.00186 | 0.00191 | 0.00199 | 0.00205 | 0.00210 |
| 430 | 0.00269 | 0.00278 | 0.00286 | 0.00298 | 0.00307 | 0.00316 | 0.00180 | 0.00186 | 0.00191 | 0.00200 | 0.00206 | 0.00212 |
| 440 | 0.00266 | 0.00275 | 0.00283 | 0.00297 | 0.00306 | 0.00315 | 0.00178 | 0.00185 | 0.00190 | 0.00199 | 0.00206 | 0.00212 |
| 450 | 0.00264 | 0.00274 | 0.00282 | 0.00296 | 0.00306 | 0.00315 | 0.00178 | 0.00184 | 0.00190 | 0.00199 | 0.00206 | 0.00213 |
| 460 | 0.00257 | 0.00267 | 0.00275 | 0.00289 | 0.00299 | 0.00309 | 0.00173 | 0.00180 | 0.00186 | 0.00195 | 0.00202 | 0.00209 |
| 470 | 0.00255 | 0.00265 | 0.00274 | 0.00288 | 0.00299 | 0.00309 | 0.00173 | 0.00180 | 0.00186 | 0.00195 | 0.00203 | 0.00210 |
| 480 | 0.00243 | 0.00252 | 0.00261 | 0.00275 | 0.00286 | 0.00296 | 0.00164 | 0.00171 | 0.00177 | 0.00186 | 0.00193 | 0.00201 |
| 490 | 0.00240 | 0.00250 | 0.00259 | 0.00273 | 0.00284 | 0.00295 | 0.00163 | 0.00170 | 0.00176 | 0.00186 | 0.00193 | 0.00201 |
| 500 | 0.00226 | 0.00236 | 0.00244 | 0.00258 | 0.00269 | 0.00280 | 0.00154 | 0.00160 | 0.00166 | 0.00176 | 0.00183 | 0.00190 |
| 510 | 0.00215 | 0.00224 | 0.00232 | 0.00246 | 0.00257 | 0.00267 | 0.00146 | 0.00152 | 0.00158 | 0.00167 | 0.00174 | 0.00181 |
| 520 | 0.00205 | 0.00214 | 0.00222 | 0.00236 | 0.00246 | 0.00257 | 0.00140 | 0.00146 | 0.00151 | 0.00161 | 0.00168 | 0.00175 |
| 530 | 0.00182 | 0.00190 | 0.00197 | 0.00210 | 0.00219 | 0.00229 | 0.00122 | 0.00128 | 0.00133 | 0.00141 | 0.00147 | 0.00154 |
| 540 | 0.00174 | 0.00182 | 0.00189 | 0.00202 | 0.00211 | 0.00221 | 0.00117 | 0.00123 | 0.00128 | 0.00136 | 0.00142 | 0.00148 |
| 550 | 0.00162 | 0.00170 | 0.00177 | 0.00188 | 0.00197 | 0.00207 | 0.00109 | 0.00114 | 0.00119 | 0.00127 | 0.00133 | 0.00139 |
| 560 | 0.00148 | 0.00155 | 0.00162 | 0.00173 | 0.00181 | 0.00190 | 0.00099 | 0.00104 | 0.00108 | 0.00115 | 0.00121 | 0.00126 |
| 570 | 0.00132 | 0.00139 | 0.00145 | 0.00155 | 0.00163 | 0.00172 | 0.00088 | 0.00093 | 0.00096 | 0.00103 | 0.00108 | 0.00113 |
| 580 | 0.00131 | 0.00137 | 0.00143 | 0.00154 | 0.00162 | 0.00170 | 0.00088 | 0.00092 | 0.00096 | 0.00103 | 0.00108 | 0.00113 |
| 590 | 0.00131 | 0.00138 | 0.00144 | 0.00155 | 0.00163 | 0.00172 | 0.00089 | 0.00094 | 0.00098 | 0.00104 | 0.00110 | 0.00115 |
| 600 | 0.00117 | 0.00124 | 0.00129 | 0.00139 | 0.00147 | 0.00155 | 0.00079 | 0.00083 | 0.00087 | 0.00093 | 0.00098 | 0.00103 |
| 610 | 0.00121 | 0.00127 | 0.00134 | 0.00144 | 0.00152 | 0.00161 | 0.00083 | 0.00087 | 0.00091 | 0.00097 | 0.00102 | 0.00108 |
| 620 | 0.00129 | 0.00137 | 0.00144 | 0.00155 | 0.00164 | 0.00174 | 0.00090 | 0.00095 | 0.00099 | 0.00107 | 0.00112 | 0.00118 |
| 630 | 0.00135 | 0.00143 | 0.00150 | 0.00162 | 0.00172 | 0.00183 | 0.00095 | 0.00101 | 0.00105 | 0.00113 | 0.00119 | 0.00126 |
| 640 | 0.00142 | 0.00150 | 0.00158 | 0.00172 | 0.00182 | 0.00194 | 0.00102 | 0.00108 | 0.00113 | 0.00121 | 0.00128 | 0.00135 |
| 650 | 0.00147 | 0.00157 | 0.00165 | 0.00179 | 0.00191 | 0.00203 | 0.00108 | 0.00114 | 0.00119 | 0.00128 | 0.00135 | 0.00143 |
| 660 | 0.00151 | 0.00161 | 0.00170 | 0.00185 | 0.00197 | 0.00210 | 0.00112 | 0.00118 | 0.00124 | 0.00134 | 0.00141 | 0.00149 |
| 670 | 0.00156 | 0.00166 | 0.00175 | 0.00191 | 0.00204 | 0.00218 | 0.00116 | 0.00123 | 0.00129 | 0.00139 | 0.00147 | 0.00156 |
| 680 | 0.00159 | 0.00169 | 0.00179 | 0.00196 | 0.00209 | 0.00224 | 0.00119 | 0.00126 | 0.00133 | 0.00144 | 0.00152 | 0.00161 |
| 690 | 0.00160 | 0.00171 | 0.00181 | 0.00198 | 0.00212 | 0.00227 | 0.00121 | 0.00128 | 0.00135 | 0.00146 | 0.00155 | 0.00164 |
| 700 | 0.00161 | 0.00173 | 0.00183 | 0.00201 | 0.00215 | 0.00231 | 0.00123 | 0.00130 | 0.00137 | 0.00149 | 0.00158 | 0.00168 |






**Figures**

**a)**

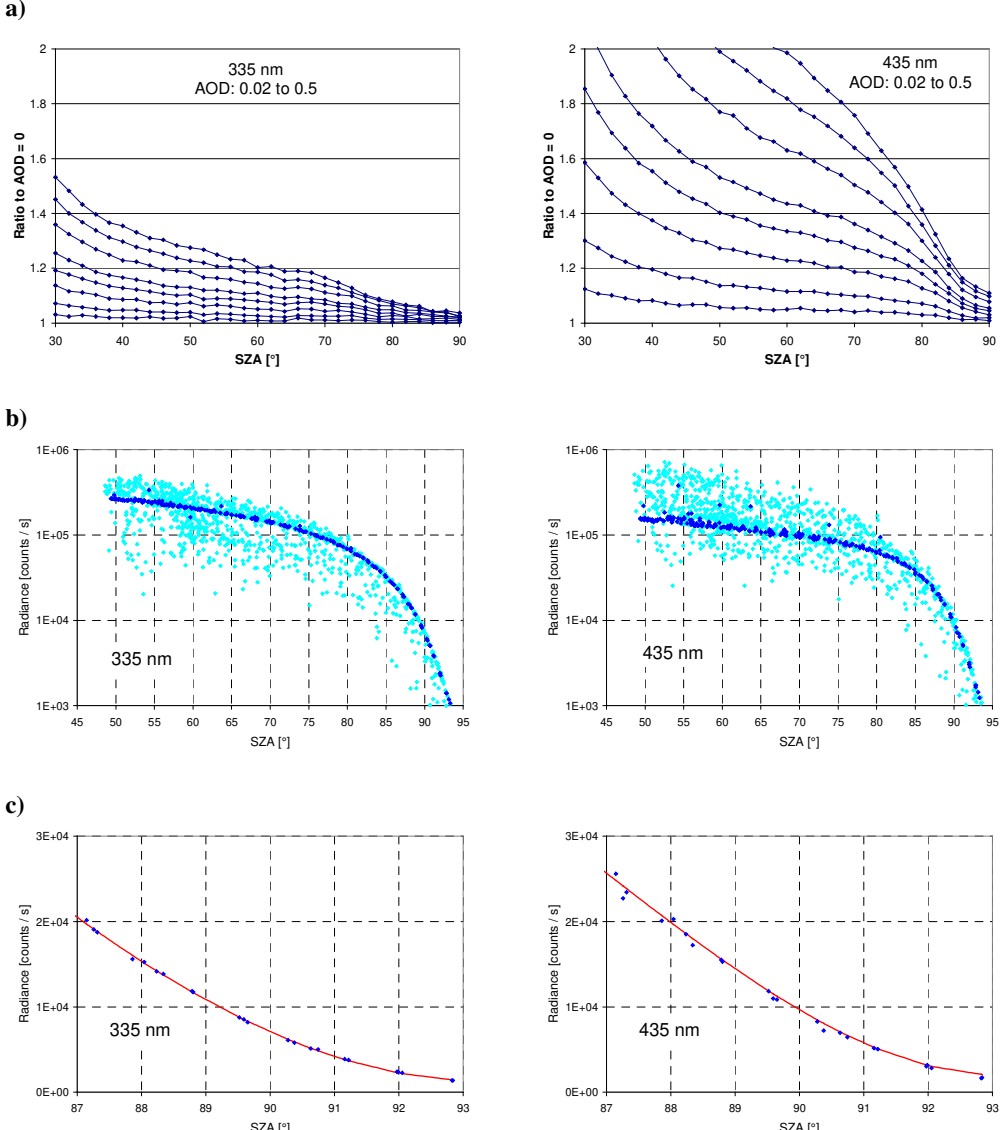

**b)**

**c)**

**Fig. 1 a) Ratio of the simulated zenith radiance for different aerosol optical depths to the radiance for an atmosphere without aerosols as function of the SZA for 335 nm (left) and 435 nm (right). Data are taken from Fig. 5 in Wagner et al. (2015). In that study simulations were only performed for SZA up to 90°. AOD values are 0.02, 0.05, 0.1, 0.15, 0.2, 0.3, 0.4, 0.5. At 90° SZA, the variation of the radiance is smallest (about 4 % at 335 nm and 11 % at 435 nm). b) measured zenith radiances from 19 September to 18 October 2022 in Mainz, Germany, for all sky conditions (light**

**blue) and clear sky conditions (dark blue). c) radiances during cloud-free conditions together with a fitted polynomial (for SZA between 85° and 93°). Note that in Fig. 1b a logarithmic y-axis is used, because the data span more than two orders of magnitude.**


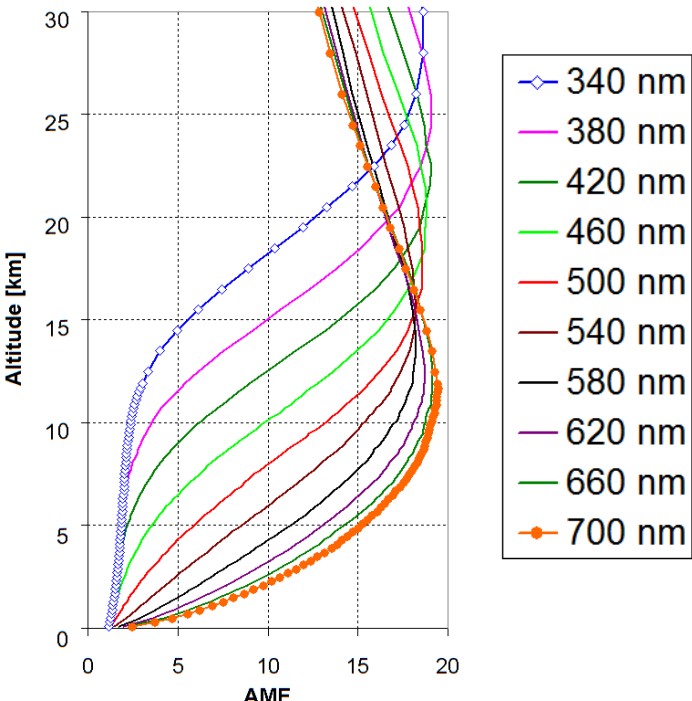

**Fig. 2 Altitude dependence of the box-AMFs for observations of zenith-scattered sun light during twilight (averages for simulations at SZA of 89° and 90°). Box-AMFs close to unity indicate an almost vertical light path. Simulations are performed for an atmosphere without aerosols.**

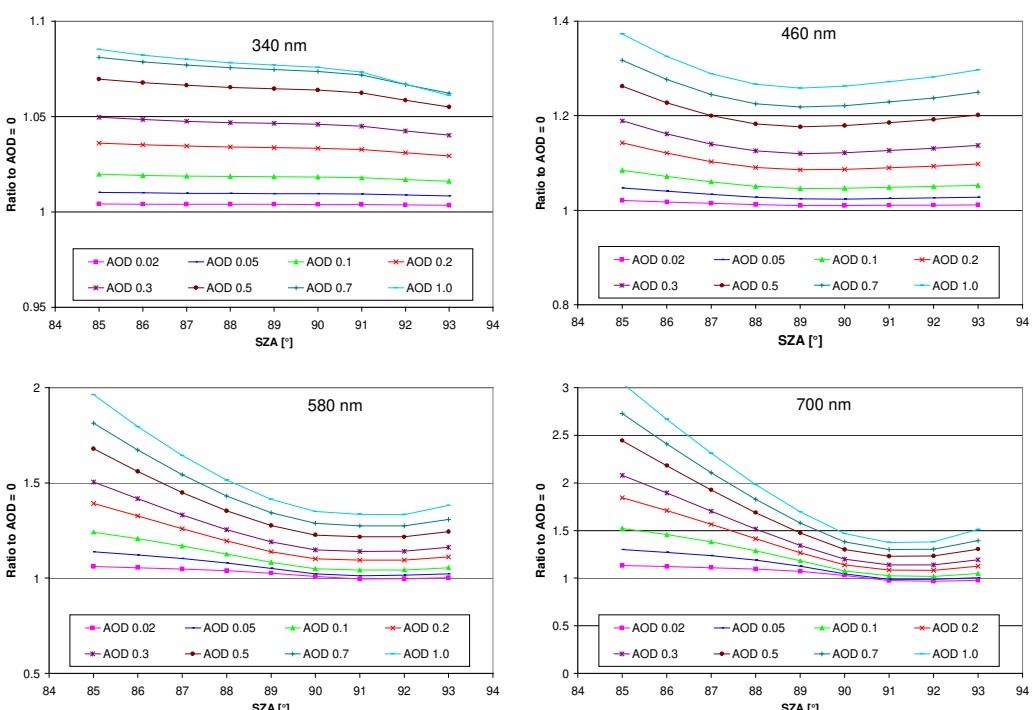

**Fig. 3 Ratio of the simulated radiances for different AODs compared to the simulation results without aerosols as**
**function of the SZA for the standard scenario, see Table 1. Results for additional wavelengths are shown in Fig. A1 in the appendix. Note the different y-axes.**

AOD = 0.1                                             AOD = 0.3

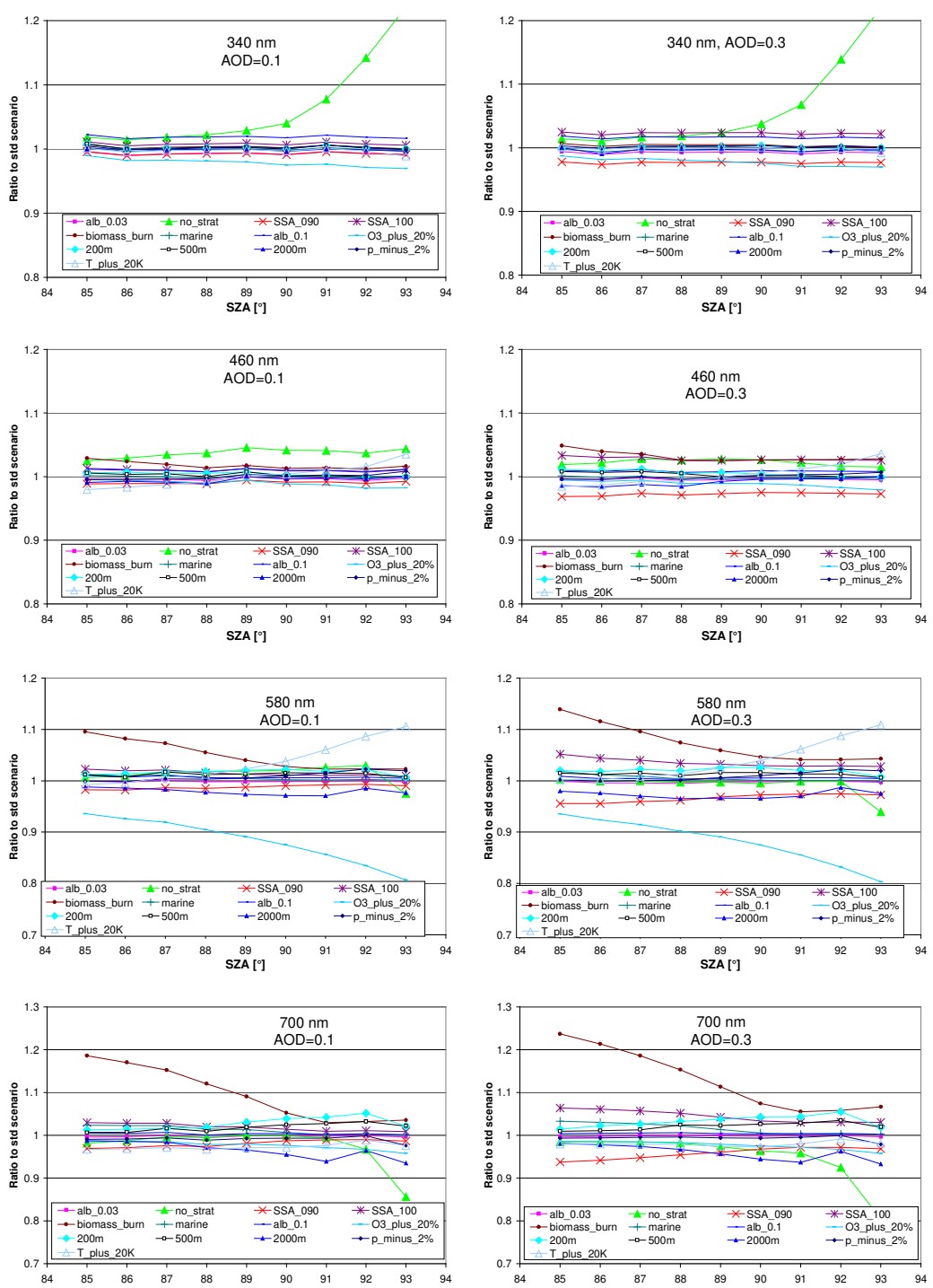

**Fig. 4 Ratio of the simulated radiance for different aerosol properties and further input parameters compared to the radiances for the corresponding standard scenarios for AOD of 0.1 (left) and 0.3 (right) as function of the SZA. Note the different y-axes. Results for additional wavelengths are shown in Fig. A3 in the appendix.**

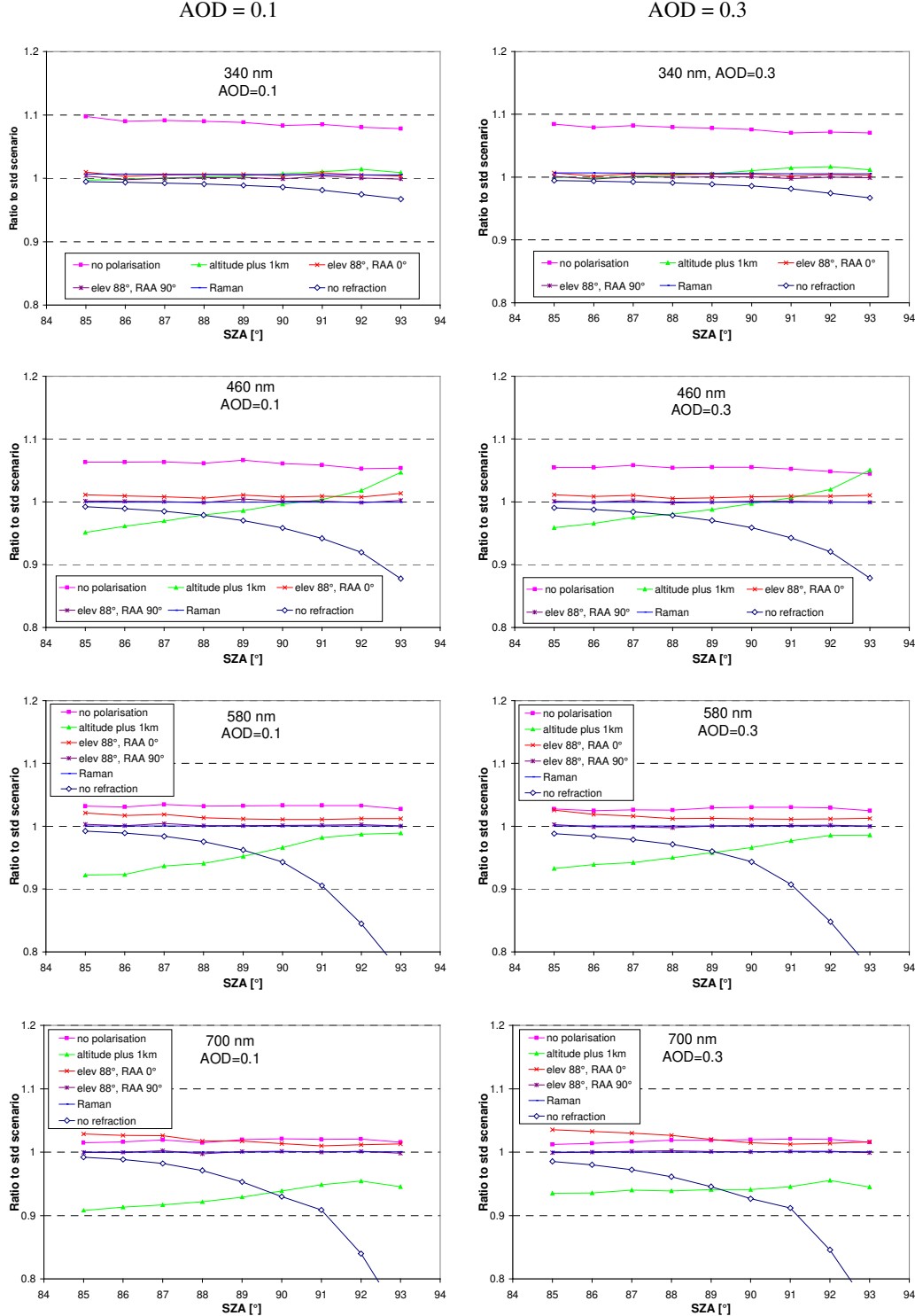

**Fig. 5 Ratio of the simulated radiances for different instrumental properties and chosen options of the radiative transfer simulations compared to the radiances of the corresponding standard scenarios for AOD of 0.1 (left) and 0.3 (right) as function of the SZA. Note the different y-axes. Results for additional wavelengths are shown in Fig. A4 in the appendix.**

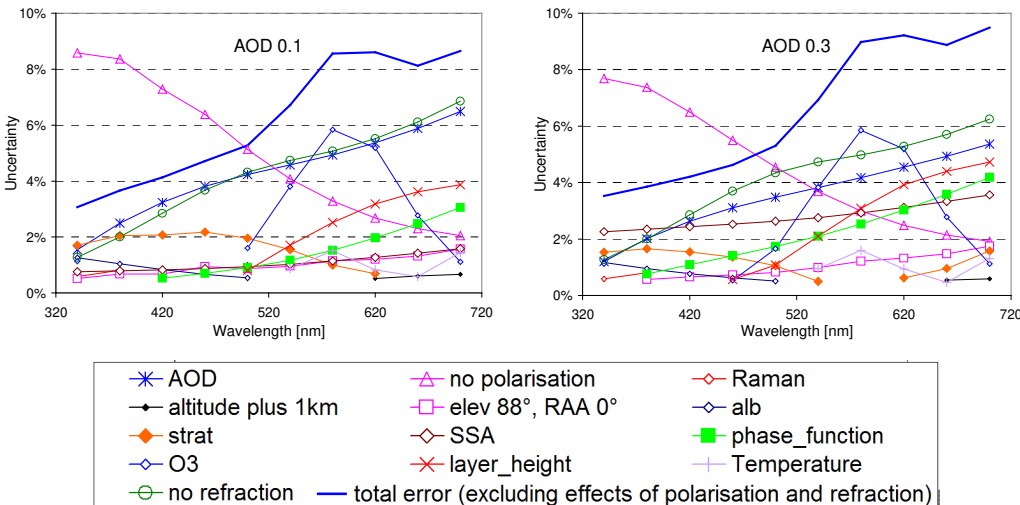

**Fig. 6 Total and individual errors for the scenario with AOD of 0.1 (left) and 0.3 (right) for zenith measurements during twilight (SZA between 89° and 90°). To enhance clarity, values below 0.5% are not shown.**


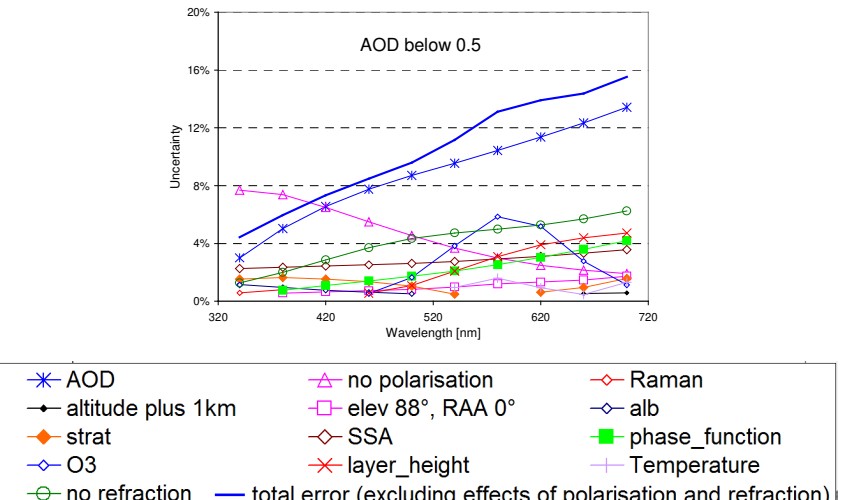

**Fig. 7 Total and individual errors for the scenario with AOD of 0.25 ±0.125 for zenith measurements during twilight (SZA between 89° and 90°). To enhance clarity, values below 0.5% are not shown.**



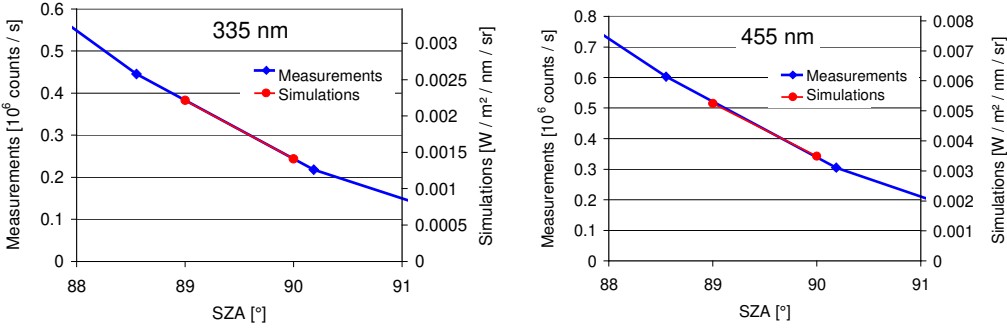

**Fig. 8 Examples of the determination of the calibration factors for two selected wavelengths. The linearly interpolated measurement results (blue) are compared to the linearly interpolated simulation results for the standard scenario (red).**

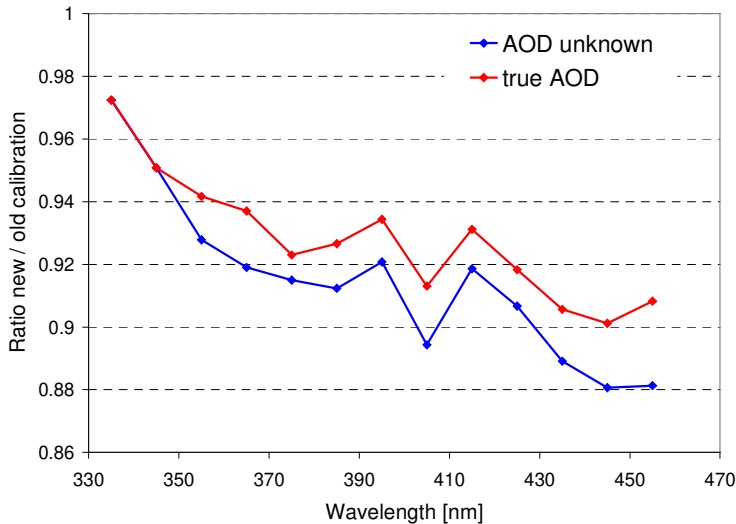

**Fig. 9 Ratio of the calibration factors for the scenarios with unknown AOD (blue) and known AOD (red) versus the calibration factors of the original study (values <1 mean that the new calibration leads to higher radiances).**

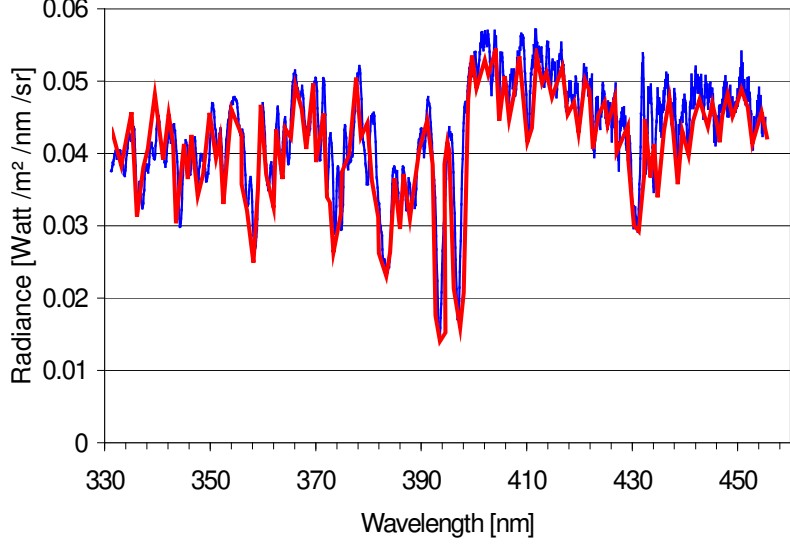


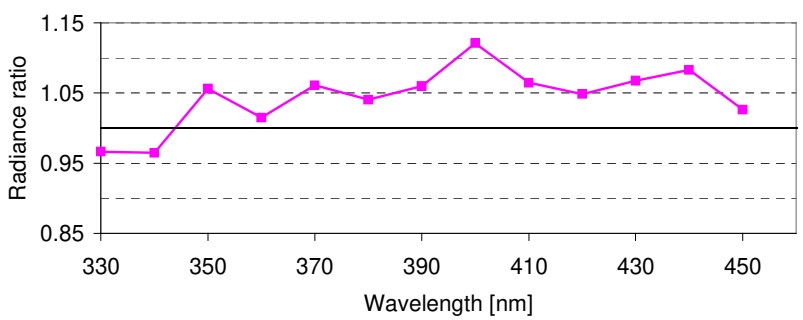

**Fig. 10 Top: Comparison of a calibrated spectrum (blue), measured on 24 June 2009, 6:54, at a SZA of 61° to an independent measurement under similar conditions (red) on 2 May, 2007 in Hannover, Germany (clear sky, SZA: 62°, Seckmeyer et al., 2009). The measurement in Hannover was scaled by a factor of 0.985 to account for the effect of the slightly different viewing geometries: The measurements in Hannover were made at exact zenith view and a SZA of 62°, while our measurements were made at 85° elevation and SZA of 61° (the figure is similar to Fig. 9 in Wagner et al., 2015). Bottom: ratio of both measurements (10 nm averages of the MAX-DOAS measurement are divided by the corresponding averages of the reference measurement in Hannover).**





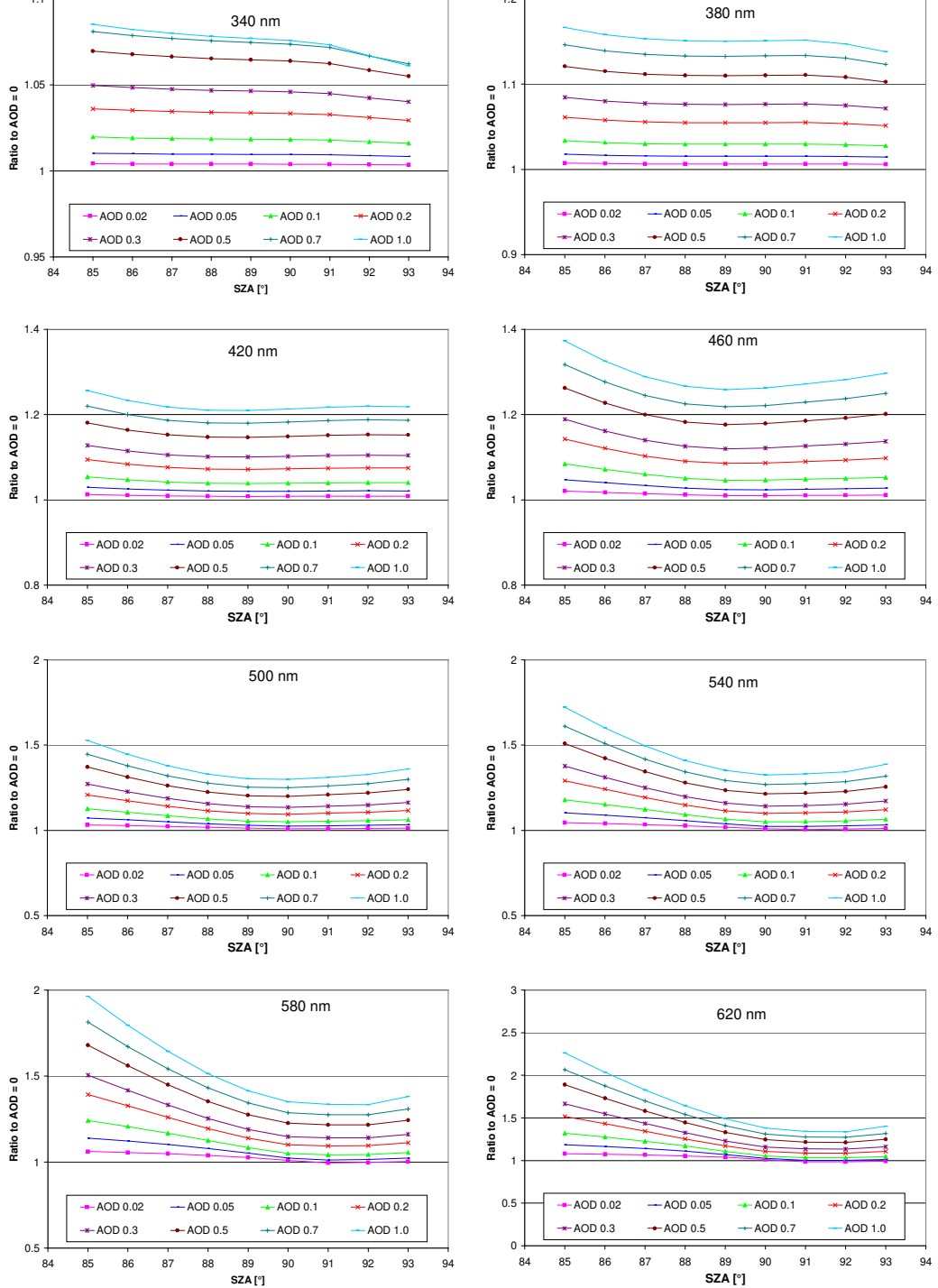

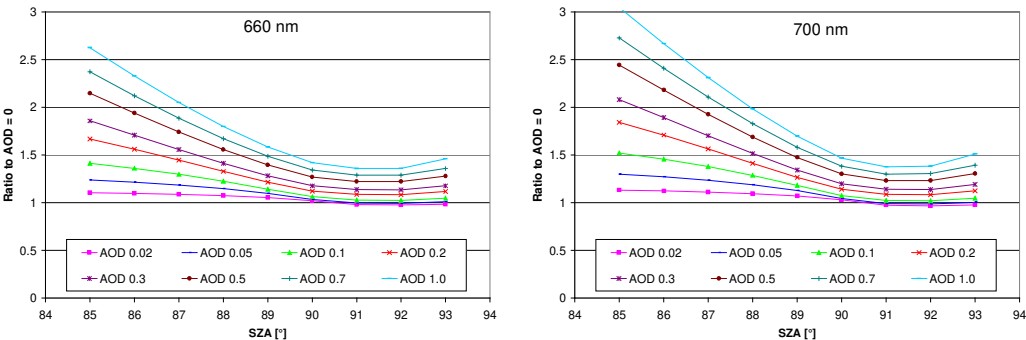

**Fig. A1 Ratio of the simulated radiances for different AODs compared to the simulation results without aerosols as function of the SZA for the standard scenario, see Table 1. Note the different y-axes.**

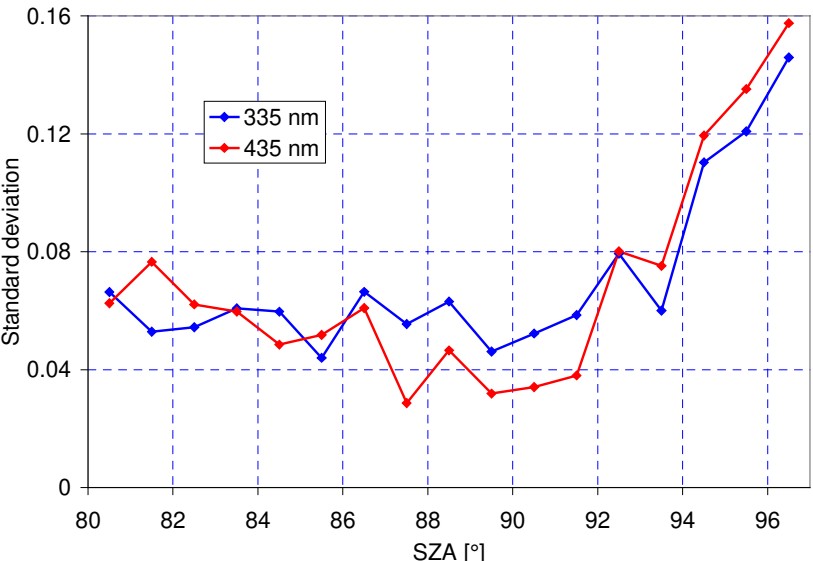

**Fig. A2 Variation of the measured radiances on 36 mostly clear days during February and March 2021 (pre-Monsun season) at Greater Noida (India). Shown are the standard deviations within 1° bins after the SZA-dependence was removed by a fitted polynomial. During the measurement period the AOD (at 550 nm, observed by MODIS) varied between 0.3 and 1.7. The measurements were carried out by Manish Sharma at Sharda University.**




AOD = 0.1                                    AOD = 0.3

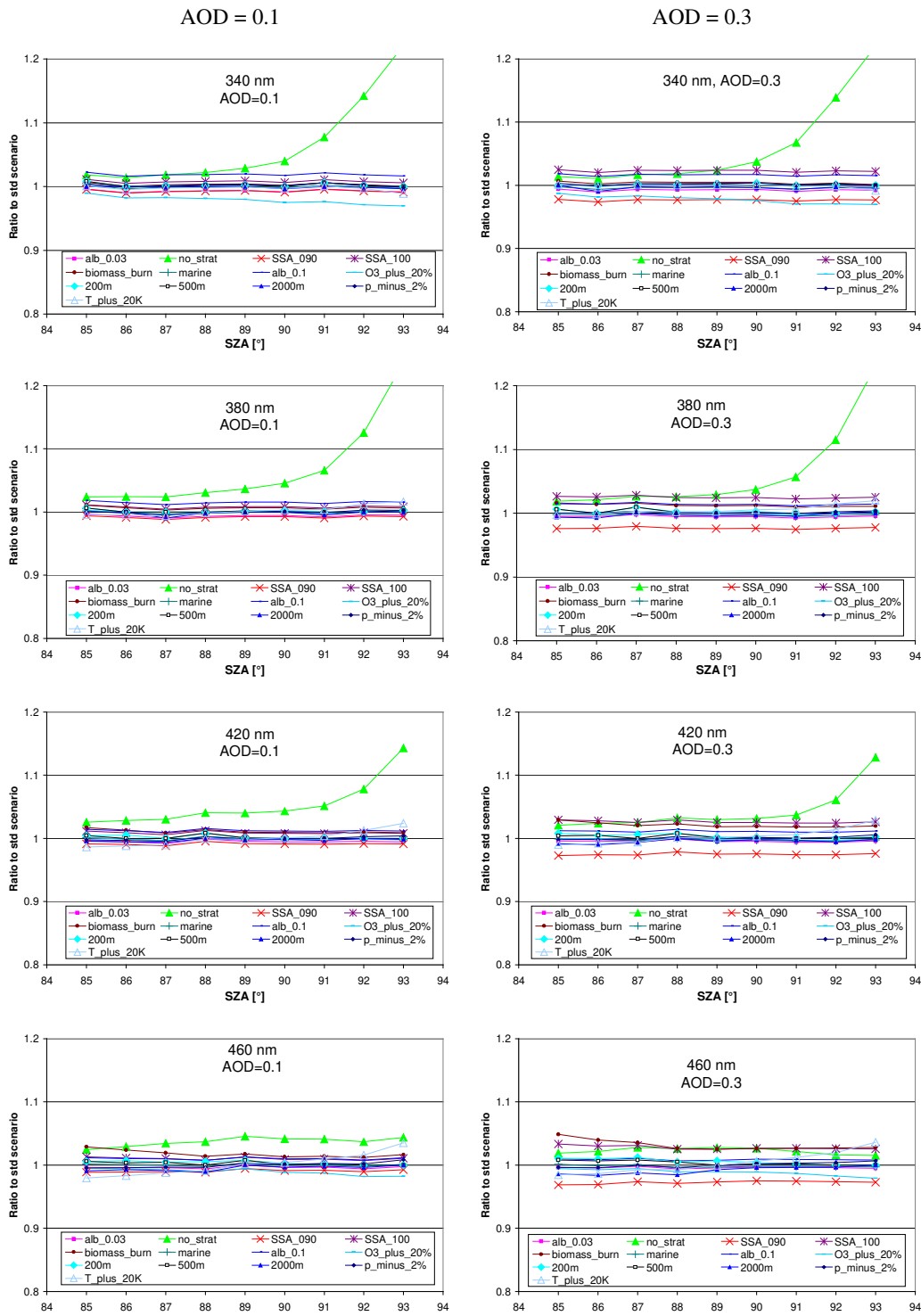

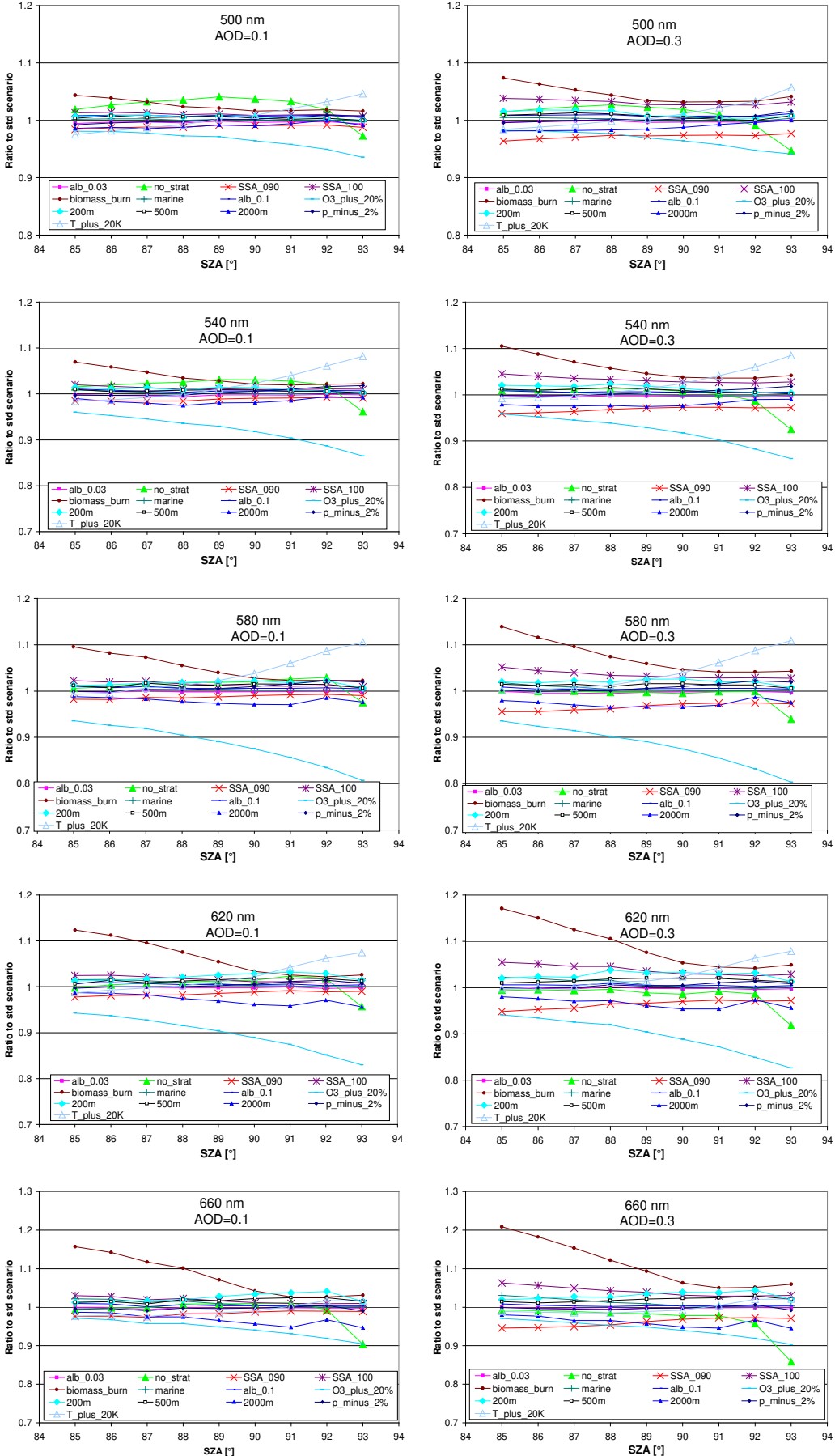

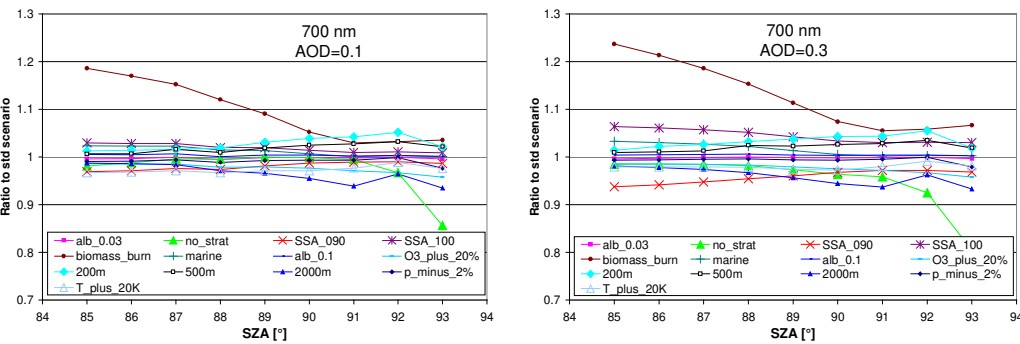

**Fig. A3 Ratio of the simulated radiance for different aerosol properties and further input parameters compared to the radiances for the corresponding standard scenarios for AOD of 0.1 (left) and 0.3 (right) as function of the SZA. Note the different y-axes.**



AOD = 0.1                                      AOD = 0.3

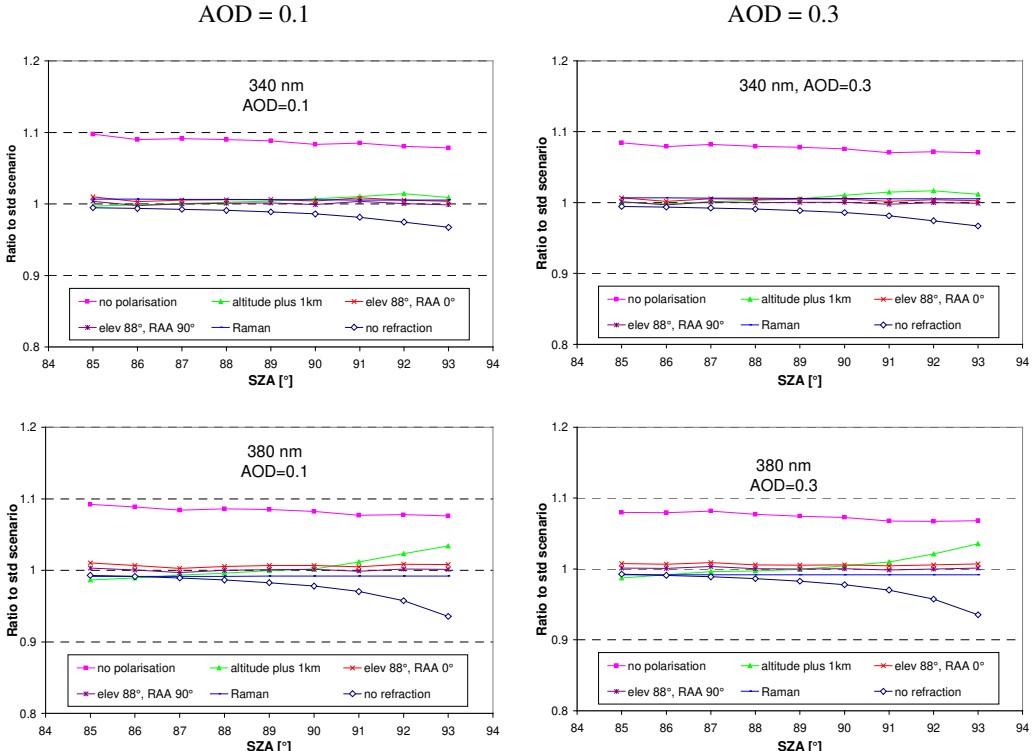

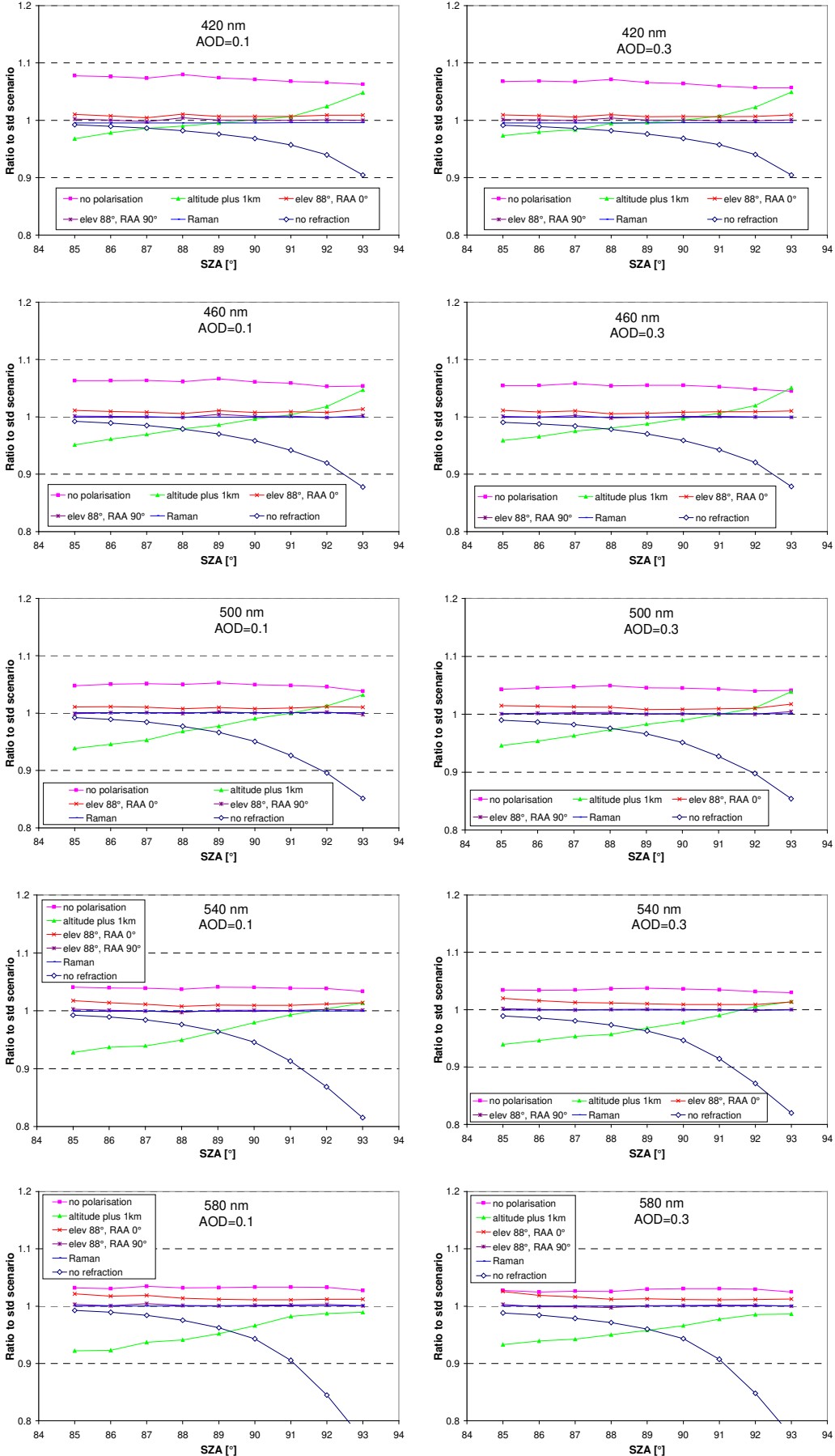

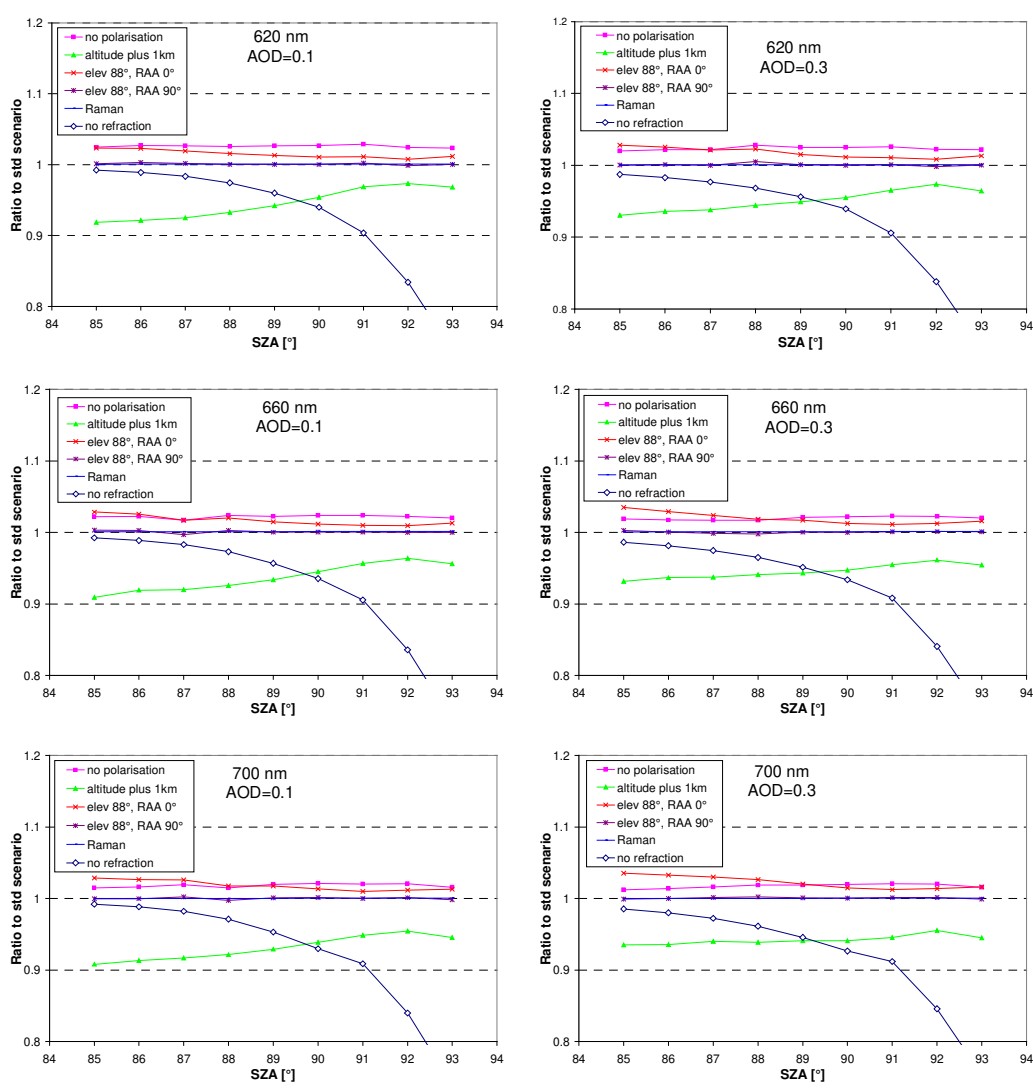

**Fig. A4 Ratio of the simulated radiances for different instrumental properties and chosen options of the radiative transfer simulations compared to the radiances of the corresponding standard scenarios for AOD of 0.1 (left) and 0.3 (right) as function of the SZA.Note the different y-axes.**
