# Peer review of "Absolute radiance calibration in the UV and visible spectral range using atmospheric observations during twilight"

_Atmospheric Measurement Techniques, 2023_

## Author Response (AR1)

Dear Joanna,

we have uploaded our responses to both reviewers. We addressed almost all points as detailed in the responses to both reviewers. Our answers are also copied below into this letter.

With best regards,

Thomas
* * *
Reply to comments from reviewer #1

The reviewed paper, "Absolute radiance calibration in the UV and visible spectral range using atmospheric observations during twilight", by Thomas Wagner, and Jānis Puķīte, presents an extension of the radiance calibration method introduced by Wagner et al. (2015). The proposed streamlined approach offers multiple benefits: the rapid (minutes) sequence of observations, less sensitivity to the pointing accuracy and aerosol load, besides the initially offered 'portability' of the otherwise cumbersome absolute calibration routines. The approach is quite sound in theory, however pending extensive validation.

Author reply:
We thank the reviewer for the positive assessment and the good suggestions. We followed almost all of them as detailed below.

Specific comments.
Considering this manuscript as a potential practical guide, I suggest the following clarifications and supplements:
Fig. 1 b & c: the form of the fitted blue-dot rad-SZA dependence seems to be different (convex vs. concave). Is this the plotting range or something else?

Author reply:
This is simply an effect of the different y-axes scaling (linear or logarthmic). A logarithmic scale was usede in Fig. 1b, because the data span several orders of magnitude, while Fig. 1c only covers a small SZA range, for which a linear scale can be used. We added a corresponding explanation to the figure caption.

Section 2.1. Mention how SCIATRAN accounts for the atmospheric refraction – this is especially relevant for the proposed calibration routine.

Author reply:
We are very thankful for this comment, because in the simulations for the original version of the paper, we did not account for atmospheric refraction. We now investigated the effect of atmospheric refraction and added the corresponding results to Figs. 5 and A4. We found that especially for long wavelengths the effect of refraction can become rather large (up to several percent for SZA of 89° and 90°, for larger SZA, it can become even larger). As a consequence, we included refraction in

the simulations, which were used for the claibration of the measured spectra. We also replaced the normalised radiances in table 3 with the corresponding simulation results considering refraction. We also explicitely state in the conclusions that for the application of the method, atmospheric refraction has to be taken into account.

In the paper, we added more information about the radiative transfer models (and also information requested by the other reviewer), and changed the first part of section 2.1 to:

In the original study, the radiative transfer model MCARTIM-3 (Deutschmann et al., 2011) was used. MCARTIM is a a full spherical Monte-Carlo model, for which, however, under twilight conditions the noise can become rather high. Thus, in this study we decided to use the radiative transfer model SCIATRAN (version 3.8.11, Rozanov et al., 2017; Mei et al., 2023, https://www.iup.uni-bremen.de/sciatran/) in order to minimise the uncertainties due to noise of the simulations and the computational effort. We use SCIATRAN in sphericity mode and considered polarisation. For the comparison to measurements (section 3), also rotational Raman scattering and atmospheric refraction is considered (in the ray tracing calculations). A Lambertian surface albedo was used. For the conditions of this paper, the zenith scattered radiances during twilight, simulated by both models agree within ±1%. Other radiative transfer models with similar capabilities could also be used for the radiance calibration.

Also, I suggest mentioning the instrument characteristics (e.g., FOV, spectral resolution, etc.) that are accounted for in the simulations.

Author reply:
We added the following information at the beginning of section 3:

The validation of the method using measured data is difficult, because without dedicated campaigns, reference data sets at the same location as the DOAS measurements are usually not available. Thus, like in the original study (Wagner et al., 2015), we apply the method to the MAX-DOAS measurements made with the MPIC instrument during the CINDI (I) campaign in Cabauw, the Netherlands, during the morning of 24 June 2009 (Piters et al., 2012). This procedure enables a direct comparison of the calibration results of the original and refined method. The MPIC MAX-DOAS instrument is a so-called Mini-MAX-DOAS instrument which covers the spectral range from 312 to 458 nm with a spectral resolution between 0.45 and 0.8. Its field of view is about 1.2°. The spectral characteristics of the instrument were exactly taken into account in the calibration procedure as described in detail in Wagner et al. (2015).

Since the surface albedo is included in the simulations, please describe what approximation is used: a Lambertian surface or anything else?

Author reply:
We added the information to section 2.1. that we used a Lambertian albedo.

Section 2.2. For practical purposes, the finding that aerosols have the smallest impact at SZA=89-90 deg is important. However, this is yet to be proved by observations. Is

there any possibility of providing additional support from the available observations? This conclusion is not very evident while perusing the data in Fig. 1b.

Author reply:
We added a new Figure (new Figure A2) in the appendix. It shows results from MAX-DOAS measurements made in India during the pre-Monsun season in 2021. During that period, clear sky conditions were present for most of the days. Fig. A2 shows the standard deviation (in 1° degree bins) of the measured radiances for that period as function of the SZA for the two wavelengths (335nm and 355 nm) shown in Fig. 1. It is found that around 90°, the smalles variability is found. The minimum is more pronounced at 435 nm, while at 335 nm the variability only slighty increases towards smaller SZA (consistent with the results shown in Fig. 3).

At the end of section 2.2, we added the information, that also for measurements the minimum variability is found around 90° (see new Figure A2).

Table 1. Please describe how you vary the standard temperature, pressure and O3 profiles: are these simple multiplicative factors (e.g., applied to all altitudes, in %), or additive ones (selected altitudes)?

Author reply:
The changes of the ozone profile and pressure profile were multiplicative; the changes of the temperature profile were additive. The profiles were changed over the whole altitude range. This information was added to table 1.

Fig. 4. If 'alb_0.1' stands for the surface albedo, then Table 2 should include the '0.1' case.

Author reply:
Many thanks for this hint! 0.1 is correct. The entries in table 2 and 3 were changed accordingly.

Sect. 2.4. The instrument elevation is an important factor, decisively impacting the simulated radiances (Fig. 5). However, it is not clear how alt=1km case accounts for the altitude of the aerosol layer, considering that its default value is also 1km (Table 1). Please clarify.

Author reply:
In table 1 the information is added that the aerosol layer is also shifted by 1000m.

Table 2 and the text. The AOD uncertainties dominate the total error budget. However, there is no related description in Table 2. Please provide the entry and describe the approach in some detail, considering the importance of this component.

Author reply:
The corresponding information was added to the table.

l.168 and Fig. 8: The case of SZA=89.6 deg is not shown in Fig. 8, so please add it there, unless there is a typo (89.6 instead of the correct 88.6). If there is no typo, then the used SZAs, 89.6 and 90.2 deg, could be (?) too close for a safe extrapolation to

89.0 deg. The shown SZA=88.6 deg observation may be a better choice. On the other hand, if the shown red line is an interpolation between sza=88.6 and sza=90.2, then why does it deviate (if slightly) from the blue line in the 455 nm plot?

Author reply:
Many thanks for that hint! 88.6° is correct, and the text in line 168 is changed accordingly. The red line is an interpolation between the simulations at 89° and 90° SZA. It deviates slightly from the interpolated measurements at 88.6° and 90.2° (in blue). Note that in the revised version of the manuscript, simulation results taking into account atmospheric refraction are shown in this figure (in the original version simulations without refraction were shown).

Table 3. Please remind the reader how the tabulated radiances are normalized and mention the units.

Author reply:
The units (simulated radiance / solar irradiance) was added to the table.

Fig. 9. Is there any explanation for the wavelength dependence of the shown ratios?

Author reply:
We added the following possible explanation at the end of section 3:

c) For smaller SZA (like used in the original study) the effect of uncertainties in the scattering phase function is larger than for SZA close to 90°. And this source of uncertainty increases with wavelength. This might at least partly explain the increasing deviation between both calibrations with increasing wavelengths.

l.208. The shown (Fig.10) comparison could be an important validation step, besides the very few provided in the paper. The authors state that: 'Overall, good agreement is found.' This must be proven unless the authors want to dismiss this step altogether. I suggest adding to Fig. 10 a panel that shows obs.-calc. difference, in %. There seems to be some wavelength dependence in the [yet to be shown] obs.-calc. difference. If true, then can the simulations be adjusted (e.g., the starting assumptions about AOD) to minimize this wavelength dependence?

Author reply:
A panel with the relative differences (for 10 nm averages) was added to Fig. 10. The average difference is about +5% (the calibrated MAX-DOAS spectrum is higher than the reference measurement in Hanover.
The text in section 3.1. was modified to:

Figure 10 presents a comparison of the calibrated radiance spectrum measured on 24 June 2009 at 6:54 at a SZA=61° (blue) and a radiance spectrum measured in zenith direction under similar atmospheric conditions (clear sky, SZA = 62°) on 2 May 2007 in Hanover, Germany. We selected this measurement for comparison, because we found no better suited example for the validation of our method in the scientific literature. This reference spectrum was measured by an instrument specifically designed for atmospheric radiance measurements (Wuttke et al., 2006; Seckmeyer et al., 2009), and was calibrated using a calibration light source. The instrument took

part in international comparison studies and was shown to comply with NDSC Standards (Wuttke et al., 2006). The sun-earth distances were quite similar for both measurements. Although the observation geometries and atmospheric conditions are similar for both measurements, still slight deviations can be expected because of the slightly different atmospheric aerosol load. Unfortunately, there was no sun photometer observation available directly at the Hannover measurement site, but from the AERONET station in Hamburg (about 130 km north of Hannover) a slightly lower AOD compared to Cabauw was found (0.13 compared to 0.17 at 360 nm; 0.10 compared to 0.12 at 440 nm). This difference could explain about 2% to 3% higher radiances in Cabauw compared to Hannover. Note that the measurement in Hannover was scaled by a factor of 0.985 to account for the effect of the slightly different viewing geometries (exact zenith viewand SZA of 62°, compared to 85° elevation angle and SZA of 61° of our measurement, see Wagner et al., 2015). The bottom of Fig. 10 shows the ratio of both measurements (after the radiances were averaged over intervals of 10 nm). Overall, good agreement is found with the measurements at Cabauw on average about 5% higher. About half of this difference can be attributed to the different aerosol loads as described above. Part of the deviations (especially for the high frequency structures) are probably also related to the fact that the values of the reference spectrum from Hannover were manually extracted from the figure in Seckmeyer et al. (2009), because the spectral data were not available.

Please note that the spectral range of the figure was canged to 330 – 460 (the range for which the updated method can be applied).
* * *
Reply to comments from reviewer #2

The manuscript of Wagner and Pukite describes an innovative new in-field radiance calibration method utilizing radiance measurements during twilight conditions together with radiative transfer simulations. The manuscript is concise and easy to follow and the methods are state-of-the-art and mostly well described. I recommend publication with minor revisions after considering the following points:

Author reply:
We thank the reviewer for the positive assessment and the many suggestions. We followed almost all of them as detailed below.

**Minor points**

Additional explanations or descriptions would be desirable at some places:

- In Section 2.1, the differences between MCARTIM-3 and SCIATRAN should be explained in more detail. It should be mentioned, for example, that MCARTIM is a Monte Carlo model and therefore may suffer from noise in low radiance conditions. Other features of SCIATRAN relevant for this study seem to be: Spherical rather than plane parallel geometry (I assume that this is important at twilight), polarized radiation, Raman scattering. How important is

SCIATRAN offering these features for this study? Could other models be used as well?

Author reply:
We added this information (and also information requested by the other reviewer), and changed the first part of section 2.1 to:

In the original study, the radiative transfer model MCARTIM-3 (Deutschmann et al., 2011) was used. MCARTIM is a a full spherical Monte-Carlo model, for which, however, under twilight conditions the noise can become rather high. Thus, in this study we decided to use the radiative transfer model SCIATRAN (version 3.8.11, Rozanov et al., 2017; Mei et al., 2023, https://www.iup.uni-bremen.de/sciatran/) in order to minimise the uncertainties due to noise of the simulations and the computational effort. We use SCIATRAN in sphericity mode and considered polarisation. For the comparison to measurements (section 3), also rotational Raman scattering and atmospheric refraction is considered (in the ray tracing calculations). A Lambertian surface albedo was used. For the conditions of this paper, the zenith scattered radiances during twilight, simulated by both models agree within ±1%. Other radiative transfer models with similar capabilities could also be used for the radiance calibration.

- The abstract should explain for which type of measurements the method is (particularly) valuable. There are many radiation instruments on the market but the study seems to primarily address MAX-DOAS or similar spectrometers. I doubt, for example, that it would be of any value for a pyrheliometer, a pyranometer or a sunphotometer measuring either total radiances or radiances in broad wavelength bands.

Author reply:
We changed the first sentence of the abstract to:
We present an improved radiance calibration method for UV / visible spectroscopic instrument s with a narrow field of view (up to a few degrees) based on the calibration method by Wagner et al. (2015).

- The introduction mentions the importance of absolute radiance calibrations, but it doesn't formulate any target uncertainty. How do uncertainties in radiance calibration affect MAX-DOAS measurements? What uncertainty should ideally be achieved?

Author reply:
We find it difficult to present explicit target uncertainties. These will also depend on the specific application. To give some information about the relationship between the uncertainty of the measured radiance and the derived quantites, we modified the first part of the introduction to:

Measurements of the atmospheric radiance are important for many applications, e.g. atmospheric remote sensing, studies of atmospheric photochemistry, optimisation of the energy yield of photovoltaic cells, the classification of sky conditions, determination of absorbing properties of aerosols, or the quantification of biologically relevant UV doses (for more details see e.g. Riechelmann et al., 2013, Wagner et al.,

2015). For some of these applications (e.g. atmospheric photochemistry or the energy yield of photovoltaic cells), the relative uncertainties of the radiance measurements will cause similar relative uncertainties of the derived quantities. For other quantities, however, the relationship can be non-linear: for example, a 5% error of the measured radiance can lead to errors of the derived aerosol single scattering albedo of up to 10% (Dubovik et al., 2000) or to a change of the number of detected optically thick clouds (Wagner et al., 2014; 2016) of up to 15%.

- Section 2.3: The section should first explain what type of sensitivities are investigated before presenting the results in Figures 4 and A2. The figures show results for various scenarios such as SAA between 0.9 and 1.0 (as compared to the reference scenario with SAA = 0.95), but the choices of these sensitivity ranges are not or not sufficiently explained.

Author reply:
The following text was added at the beginning of section 2.3:
In addition to the dependence on the AOD, in this section, the effect of other important properties is investigated. Here assumptions about the variability range of the different quantities were made (see table 2) which should be representative for typical atmospheric conditions. In specific cases, some of these properties might be outside the assumed ranges (e.g. for events with desert dust or biomass burning plumes). Such extreme cases should be avoided for the application of the calibration method.

- Similarly, in Section 2.4 the altitude of the instrument is varied by 1 km, but it remains unclear why the sensitivity to the altitude of the instrument is investigated at all, since this altitude should be known very accurately.

Author reply:
In table 3, simuated radiances are provided, which could be used by the reader to calibrate their own instrument. These radiances are calculated for the standard scenario (in particular for an altitude of the instrument of 0 m). If these radiances are used for an instrument at higher altitudes, it is important to know how strongly the measured radiances might differ from the values in table 3. Thus the following information was added to the text:
Of course, the altitude of the instrument is usually well known and could be exactly considered in the radiative transfer simulations. However, if for simplicity the radiances for the standard scenario given in table 3 were used, it will be useful to know, how strongly the altitude of the instrument affects the measured radiances.

- In section 3.1 measurements in Hanover are used for comparison, but there is no further description of these measurements, how they were collected, why they are suitable for comparison, or whether there was a more sophisticated radiance calibration applied to these measurements (or not).

Author reply:
More information about the measurements and more detailed discussion of the comparison results were added. The text in section 3.1 was modified to:
Figure 10 presents a comparison of the calibrated radiance spectrum measured on 24 June 2009 at 6:54 at a SZA=61° (blue) and a radiance spectrum measured in zenith

direction under similar atmospheric conditions (clear sky, SZA = 62°) on 2 May 2007 in Hanover, Germany. We selected this measurement for comparison, because we found no better suited example for the validation of our method in the scientific literature. This reference spectrum was measured by an instrument specifically designed for atmospheric radiance measurements (Wuttke et al., 2006; Seckmeyer et al., 2009), and was calibrated using a calibration light source. The instrument took part in international comparison studies and was shown to comply with NDSC Standards (Wuttke et al., 2006). The sun-earth distances were quite similar for both measurements. Although the observation geometries and atmospheric conditions are similar for both measurements, still slight deviations can be expected because of the slightly different atmospheric aerosol load. Unfortunately, there was no sun photometer observation available directly at the Hannover measurement site, but from the AERONET station in Hamburg (about 130 km north of Hannover) a slightly lower AOD compared to Cabauw was found (0.13 compared to 0.17 at 360 nm; 0.10 compared to 0.12 at 440 nm). This difference could explain about 2% to 3% higher radiances in Cabauw compared to Hannover. Note that the measurement in Hannover was scaled by a factor of 0.985 to account for the effect of the slightly different viewing geometries (exact zenith viewand SZA of 62°, compared to 85° elevation angle and SZA of 61° of our measurement, see Wagner et al., 2015). The bottom of Fig. 10 shows the ratio of both measurements (after the radiances were averaged over intervals of 10 nm). Overall, good agreement is found with the measurements at Cabauw on average about 5% higher. About half of this difference can be attributed to the different aerosol loads as described above. Part of the deviations (especially for the high frequency structures) are probably also related to the fact that the values of the reference spectrum from Hannover were manually extracted from the figure in Seckmeyer et al. (2009), because the spectral data were not available.

The introduction states that it will be shown later in the paper that "absorbing aerosols might still have a relatively strong effect". There is, however, very little material in the manuscript addressing this. It would be good to elaborate this point a bit more and possibly reiterate in the conclusions.

Author reply:
The sentence in the introduction is changed to:

(as will be shown later in section 2.3, the phase function and the single scattering albedo might still have a relatively strong effect)

In the conclusions we added the following information:
,Another limitation of the method is that especially for situations with enhanced AOD (see results for AOD of 0.3 in Fig. 4 and Fig. A3) the aerosol properties (phase function and single scattering albedo) can have a relatively strong effect. Such situations (e.g. desert dust events or biomass burning plumes) should be excluded from the application of the calibration technique.'

Section 2.4: The section title is confusing. What do you mean by "instrument calibration"? The only calibration mentioned in the text is the "elevation angle calibration".

Author reply:

We changed ‚calibration' to ‚properties'.

The advantage of the new calibration method is that measurements need to be performed only over a few minutes. On the other hand, it requires accurate knowledge of the solar zenith angle which changes rapidly at twilight. Shouldn't it therefore be mentioned that an exact registration of time and of the longitude/latitude position of the measurement is critical? Considering the potentially large drifts of computer clocks, this may not be as trivial as it may seem.

Author reply:
Many thanks for this important hint! We added the following information to the conclusions:
‚... care should be taken for the exact calculation of the SZA. Especially during twilight, small errors of the computer time and/or the latitude/longitude settings can lead to considerable errors of the SZA calculation..'

Section 3 refers to MAX-DOAS measurements collected during the CINDI campaign in Cabauw. Since many instruments were operated in parallel CINDI, it is unclear whether all these measurements were used or only those of a single instrument.

Author reply:
We clarified in the text that only the measurements of the MPIC instrument were used in this study.

The new calibration method sounds appealing, but I have two minor concerns: Isn't the problem of clouds particularly large during twilight conditions when the atmospheric path of sunlight is very long? Doesn't this strongly reduce the availability of suitable observations?

Author reply:
Many thanks for this hint! To address this potential error source, the following text was added in the conclusions:
‚By comparing the results from several days, also the potential effect of clouds far away from the measurement site (but still in the path of the direct sun light) could be identified and contaminated measurements could be removed.'

The second issue is that radiance levels are low during twilight. Is this sufficient to calibrate an instrument over the whole range of possible radiance levels? What about instrument/detector non-linearities?

Author reply:
Many thanks for this hint! We added the following text to the conclusions:
‚... care should also be taken that the saturation level of the detector during twilight is similar to that of typical measurements (at smaller SZA). This was the case for the measurements used in this study. Otherwise, non-linearities of the detector and/or the read-out electronics could lead to systematic errors.'

Almost all figures contain multiple lines that are difficult and sometimes very difficult to discern. It would be good to enhance symbol sizes and line widths.
Finally, I was wondering about the practical implications of this study.

Author reply:
In order to enhance the clarity, we modified figures 2, 4, 6, 7, and A3 (increased symbols for the most relevant curves).

Are there any plans, for example, to introduce this new method in routine (MAX-DOAS) observations or to post-process any existing data sets, e.g. within NDACC or ACTRIS?

Author reply:
Many thanks for this good suggestion! However, before the method could be applied in routine applications, a more comprehensive validation should be performed.

**Corrections/typos:**

Author reply:
All corrections were applied

- Page 3, line 97: The SZA range should probably be "89° to 90°" rather than "89° to 89°".

- Page 3, line 116: The sentence ends on two points rather than one.

- Page 4, line 140-141: The last part of the sentence ("which represent typical aerosol abundances") can be dropped as it was explained several times.

- Page 4, line 149: It should probably be "overestimate" rather than "underestimate".

- Page 4, line 156: The second 340 nm should be changed to 420 nm (I think).

- Page 4, line 163: replace "The" by "the" in "the Netherlands"

- Page 4, line 168: I think it should be 88.6° rather than 89.6°.

- Page 5, line 190: Remove parenthesis at the end of the sentence.

- Page 5, line 198: "the results" appears twice.

- Page 6, line 218: Sentence ends with two points rather than one.

---

## Author Response (AR2)

Dear Joana,

one of the reviewer suggested ,... adding a couple of panels to one of the figures (e.g., Figure 3, for the two borderline wavelengths) that compare the refraction-free and refraction-accounted cases.'

Following this suggestion, we decided to enhance the overall consistency of the paper by replacing all results of the sensitivity studies (figures 3, 4, 5, 6) with the correponding simulations results including refraction.

When comparing the updated results of figures 4 and 5 with the corresponding previous results, it turned out that we had made an important error in the calculation of the results shown in these figures (and thus also in the results of Fig. 6 and 7). Instead of calculating the ratio of the radiances of the various scenarios to the radiances of the standard scenario, we had calculated the ratios of the radiance differences (AOD of 0.3 minus AOD of zero, like in Fig, 3). For some of the scenarios, especially for the effects of polarisation, stratospheric aerosols, and ozone absorption, the corresponding changes compared to the original figures are quite large. Fortunately, the total uncertainties and the overall conclusions of the paper stay almost unchanged.

In the revised paper we replaced Figures 3, 4, 5, and 6 with the updated figures including atmospheric refraction (and for Figures 4,5, and 6 with the corrected ratios).

During the revision, we also made some minor corrections and changes as described below:

-we corrected the y-axis labels in Fig. 5 (,Ratio to AOD=0' was changed to ,Ratio to std scenario')

-in Fig. 6 and 7, values below 0.5% are not shown in order to enhance the clarity of the figures.

-for the calculation of the total error, the contribution of a wrong ozone profile was reduced (now half the deviation of the simulation with 20% change instead of the full deviation to the standard scenraio). This change was made, because:
a) otherwise in Figs. 6 and 7 the error related to the ozone absorption would dominate the total error for almost all wavelengths and scenarios, while in relatity the uncertainty of the knowledge of total ozone column (e.g. from satellite observations) is much smaller.
c) this treatment is consistent with the treatment of the other effects.
In section 2.3, 2.5.1, and the conclusions the importance of correct ozone data for the radiative transfer simulations is now emphasised.

-in Fig. 4 and Fig. A3, marine and biomass burning aerosols were reversed (the labelling was wrong in the original figures)

-in the original version, it was mentioned in section 2.1 that , For the comparison to measurements (section 3), also rotational Raman scattering and atmospheric refraction is considered (in the ray tracing calculations).'

This sentence was corrected to ‚For the sensitivity studies (section 2), also rotational Raman scattering is considered.', because in the standard scenario Raman scattering was not included.

The changes in the text are marked by the track change option.

Best regards,

Thomas